# Red blood cell mannoses as phagocytic ligands mediating both sickle cell anaemia and malaria resistance

Huan Cao[1], Aristotelis Antonopoulos[2], Sadie Henderson[3], Heather Wassall[1], John Brewin[4], Alanna Masson[5], Jenna Shepherd[1], Gabriela Konieczny[1], Bhinal Patel[2], Maria-Louise Williams[1], Adam Davie[1], Megan A. Forrester[1], Lindsay Hall[1], Beverley Minter[1], Dimitris Tampakis[6], Michael Moss[3], Charlotte Lennon[1], Wendy Pickford[1], Lars Erwig[1], Beverley Robertson[2], Anne Dell[4], Gordon D. Brown[1,7], Heather M. Wilson[1], David C. Rees[4], Stuart M. Haslam[2], J. Alexandra Rowe[8 ✉], Robert N. Barker[1,9] & Mark A. Vickers[1,3,5,9 ✉]

In both sickle cell disease and malaria, red blood cells (RBCs) are phagocytosed in the spleen, but receptor-ligand pairs mediating uptake have not been identified. Here, we report that patches of high mannose N-glycans (Man$_{5-9}$GlcNAc$_2$), expressed on diseased or oxidized RBC surfaces, bind the mannose receptor (CD206) on phagocytes to mediate clearance. We find that extravascular hemolysis in sickle cell disease correlates with high mannose glycan levels on RBCs. Furthermore, *Plasmodium falciparum*-infected RBCs expose surface mannose N-glycans, which occur at significantly higher levels on infected RBCs from sickle cell trait subjects compared to those lacking hemoglobin S. The glycans are associated with high molecular weight complexes and protease-resistant, lower molecular weight fragments containing spectrin. Recognition of surface N-linked high mannose glycans as a response to cellular stress is a molecular mechanism common to both the pathogenesis of sickle cell disease and resistance to severe malaria in sickle cell trait.

[1] School of Medicine, Medical Sciences and Nutrition, University of Aberdeen, Aberdeen, UK. [2] Department of Life Sciences, Imperial College London, London, UK. [3] Scottish National Blood Transfusion Service, Aberdeen, UK. [4] Department of Haematology, King's College Hospital, London, UK. [5] Department of Haematology, Aberdeen Royal Infirmary, Aberdeen, UK. [6] Centre for Biological Engineering, School of Mechanical, Electrical and Manufacturing Engineering, Loughborough University and Division of Cancer Studies, King's College London, London, UK. [7] Medical Medical Research Council Centre for Medical Mycology at the University of Exeter, Exeter, UK. [8] Centre for Immunity, Infection and Evolution, Institute of Immunology and Infection Research, University of Edinburgh, Edinburgh, UK. [9] These authors contributed equally: Robert N. Barker, Mark A. Vickers. ✉email: Alex.Rowe@ed.ac.uk; m.a.vickers@abdn.ac.uk

Sickle cell disease comprises a group of disorders affecting over 20 million individuals and is caused by a mutation causing an amino acid substitution (E6V) in the adult hemoglobin β chain,[1,2] so that the physiological hemoglobin (Hb) A tetramer, $\alpha_2\beta_2$, is replaced by the HbS tetramer $\alpha_2\beta^S_2$, which can form pathological polymers. Homozygosity for wild type and sickle cell alleles are referred to hereafter as HbAA and HbSS respectively. The disease is variable, with modifiers such as high levels of the fetal β hemoglobin chain, γ, resulting in $\alpha_2\gamma_2$ or $\alpha_2\beta^S\gamma$ tetramers that terminate HbS polymers and ameliorate disease. The milder disease is also associated with compound heterozygosity of the sickle cell allele with quantitative defects in α or β chains (thalassemias) or other hemoglobin chain variants like hemoglobin C (HbC).

Sickle cell disease is characterized by multi-system vasculopathy and hemolysis, which cause much morbidity and mortality, especially in Africa. The anemia has been ascribed to abnormal physical properties of diseased red blood cells (RBCs), which interfere with their transit through the splenic and hepatic vasculatures, so stimulating phagocytic uptake by tissue macrophages.[3] However, the observation that isolated macrophages take up HbSS RBCs selectively in vitro[4] indicates the presence of disease-specific ligands, which remain uncharacterized. Heterozygosity for HbS, sickle cell trait, affects over 250 million individuals and is maintained in the population by conferring protection against severe malaria. Individuals with sickle cell traits contain a mixture of hemoglobins referred to hereafter as HbAS. The mechanism underlying this protection is not fully explained, but the mutation has long been known to prevent high levels of parasitemia.[5,6] Yet under most conditions in vitro, the parasites grow equally well in HbSS RBCs compared to those with normal hemoglobin,[7,8] implying that protection is due to efficient immune clearance of infected HbSS RBCs, and again raising questions as to the identity of the ligands responsible for mediating phagocytosis. Here we show that oxidative stress induces RBCs to express high levels of surface high mannose glycans and this mechanism is active in both sickle cell disease and after infection by P. falciparum, especially in RBCs with sickle cell trait. The high mannose glycans can be recognized by the mannose receptor (CD206) expressed on macrophages and Lyve-1+ endothelial cells in the spleen to effect phagocytosis. High mannose glycans, therefore, act as damage-associated and pathogen-associated molecular patterns and are important in mediating both beneficial and pathological effects of the sickle cell allele.

## Results

### Red blood cells from patients with sickle cell disease express high mannose N-glycans on their surfaces.

We postulated these putative uptake ligands might be N-linked glycans, given the prominence of the glycocalyx on RBCs and the corresponding expression of lectins as key innate receptors on macrophages.[9] A survey for surface ligands using a panel of plant lectins identified two panel members that bound preferentially to HbSS RBC (Fig. 1a), *Galanthus nivalis* Agglutinin (GNA) and *Narcissus pseudonarcissus* Lectin (NPL). The binding was specific (Supplementary Table 1, Supplementary Figs. 1a–b, 2a) and both lectins were noted to have similar specificities for terminal mannose residues (Supplementary Fig. 1a).[10,11] Microscopy with fluorescent GNA lectin revealed discrete patches on the surfaces of HbSS, but not healthy (HbAA), RBCs (Fig. 1b, Supplementary Fig. 1c, d). Glycomic analysis using mass spectrometry showed that HbSS RBCs express N-linked high mannose glycans, hereafter high mannose glycans ($Man_{5-9}GlcNAc_2$; Fig. 1c), which are known ligands for phagocytosis by macrophages[9,12] and therefore good candidates for mediating RBC uptake. High mannose

glycans are also observed in the N-glycome profiles from HbAA RBC ghosts (Fig. 1c, Supplementary Fig. 3, Supplementary File 1). The proportions of high mannose glycans with respect to whole N-glycomes were not significantly different between sickle and healthy RBC ghosts (Supplementary Fig. 1e). The marked difference between GNA binding on the cell surface of HbAA compared to HbSS RBCs is therefore not explained by the total high mannose glycan content of the ghosts.

### RBC surface mannose correlates with extravascular hemolysis in sickle cell disease.

To assess the relevance of high mannose N-glycan display for RBC uptake in vivo, we exploited the heterogeneity of sickle cell disease arising from the interactions of HbS with other mutations in the globin loci (such as HbC, α-thalassemias, and β-thalassemias) that also protect against malaria.[13] If mannoses were phagocytic ligands in sickle cell disease, higher levels of mannose exposure should correlate with more severe anemia. Despite a similar glycomic profile, RBCs from patients who were homozygous for HbS tended to exhibit higher binding of GNA lectin, compared to RBCs from healthy individuals containing HbAA or those with sickle cell trait (HbAS) (Fig. 2a). Patients with sickle cell disease who were compound heterozygotes for HbS and either HbC or β-thalassemia, or who had HbSS but with mitigating α-thalassemia or high levels of fetal hemoglobin (HbF), tended to exhibit low to intermediate GNA lectin binding (Fig. 2a). RBCs in other anemias did not express high levels of exposed mannose residues (Supplementary Fig. 2a). The classical apoptotic marker for phagocytosis, phosphatidylserine, as measured by annexin V binding, was expressed at similar, low levels on RBCs from each of the clinical groups (Supplementary Fig. 2b), although it was highly expressed on positive control calcium ionophore treated, eryptotic RBCs (Supplementary Fig. 2c). Overall, GNA lectin binding correlated significantly with more severe anemia (Fig. 2b) and other markers of hemolysis (Supplementary Fig. 2d, e), consistent with high mannose N-glycan expression driving extravascular uptake by hepatosplenic phagocytes, which is the major mechanism of hemolysis in sickle cell disease.[14] A minor, but significant, the proportion of RBC loss in sickle cell disease is also accounted for by intravascular hemolysis.[14] However, plasma lactate dehydrogenase (LDH) levels, a marker of intravascular hemolysis,[14] did not correlate with RBC GNA lectin binding (Fig. 2c, d, e) within HbSS patients. We postulated that mannose-binding lectin might bind and opsonize sickle cells, but no significant correlations between levels of this plasma protein and hemolytic phenotypes were observed (Supplementary Fig. 4a–c). Furthermore, when we added cells washed free of plasma to macrophages, SS, but not AA, RBCs were selectively taken up and this could be inhibited by mannan (Fig. 2f), indicating the macrophages expressed a receptor that interacted directly with surface mannoses.

### Surface mannoses can be induced on healthy RBC by oxidative stress and are recognized by the mannose receptor (CD206).

High mannose N-glycans ($Man_{5-9}GlcNAc_2$) were detected in glycomic analyses of healthy (HbAA) RBCs (Fig. 1c, Supplementary Fig. 3). Furthermore, permeabilization of healthy RBCs allowed GNA lectin binding in patches that colocalized with the membrane skeleton (Supplementary Fig. 5a). Sickle cell disease is associated with intracellular oxidative stress,[15,16] so we determined whether exposing healthy RBCs to an oxidizing agent (Supplementary Fig. 5b) would alter the surface mannose exposure as assessed by GNA lectin binding (Fig. 3a). Under the experimental conditions applied, oxidation of healthy RBCs indeed resulted in binding of GNA lectin, with similar, but fewer,

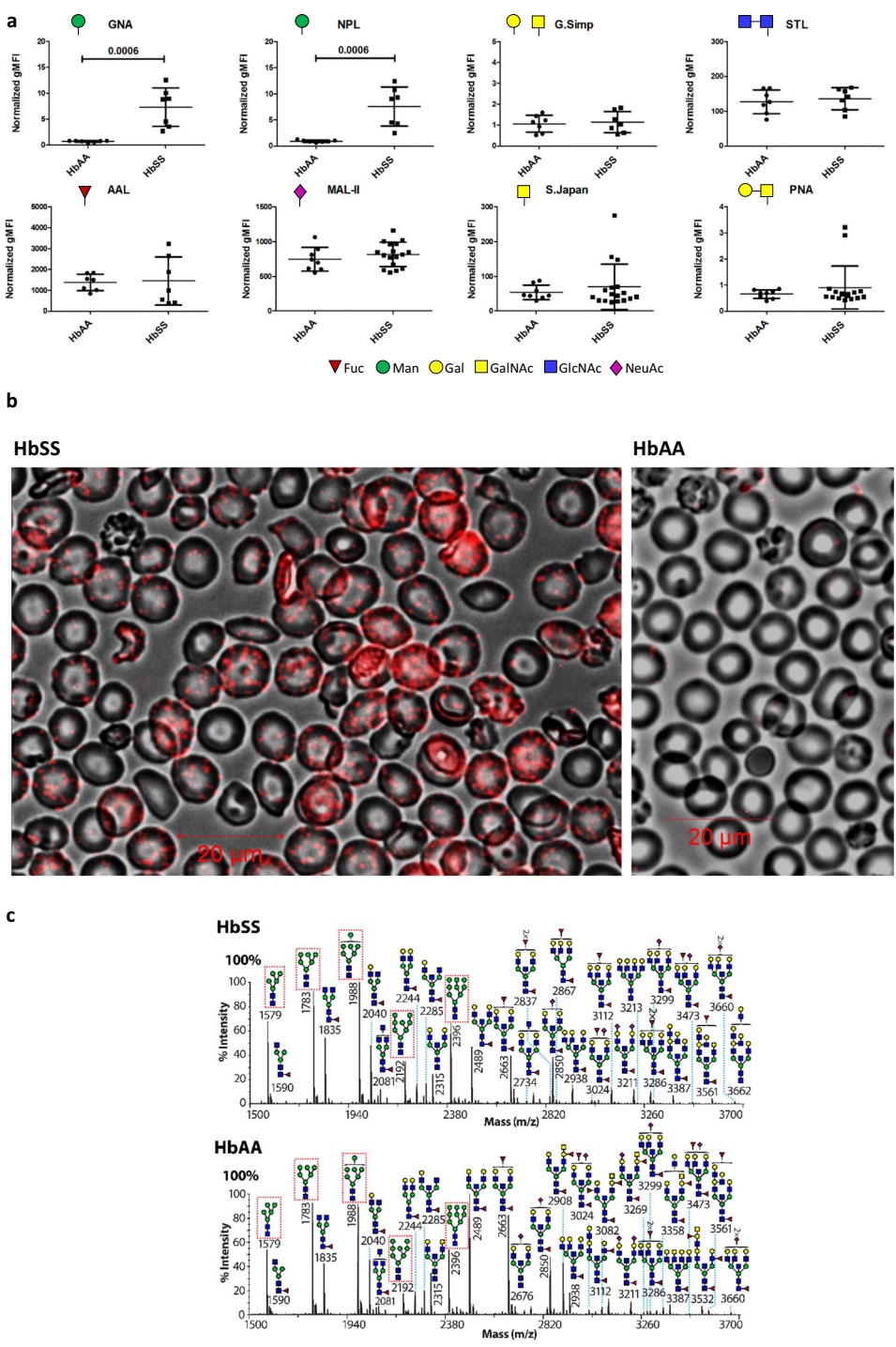

**Fig. 1 HbSS RBCs are characterized by microdomain expression of surface mannoses.** Whole blood flow cytometry analysis of HbAA (normal hemoglobin) and HbSS (homozygous sickle cell hemoglobin) RBCs using fluorescently labeled plant lectins, detailed in "Methods" section. Vertical axes show normalized geometric mean fluorescence (gMFI). Symbols of terminal carbohydrates detected by plant lectins are indicated. Data shown as median +/− IQR, n = 7 per group for significant differences, 2 tailed Mann-Whitney p values shown, distinct samples measured once each, 3 separate experiments. Annotation uses conventional symbols for carbohydrates in accordance with http://www.functionalglycomics.org guidelines: purple diamond, sialic acids; yellow circle, galactose; blue square, N-acetyl glucosamine; green circle, mannose; red triangle, fucose. **a** *Galanthus nivalis* Agglutinin (GNA) lectin staining (red) of HbSS and HbAA RBCs, immunofluorescence, merged with the bright field. **b** MALDI-ToF mass spectra (*m/z* versus relative intensity) for glycomic analysis of N-glycans from membrane ghosts from individual HbSS and HbAA donors. Red boxes indicate high mannose structures. Annotation uses conventional symbols for carbohydrates in accordance with http://www.functionalglycomics.org guidelines: purple diamond, sialic acids; yellow circle, galactose; blue square, N-acetyl glucosamine; green circle, mannose; red triangle, fucose. Only major structures are annotated for clarity. Full spectra from both HbSS and HbAA donors are shown in Supplementary Fig. 3c.

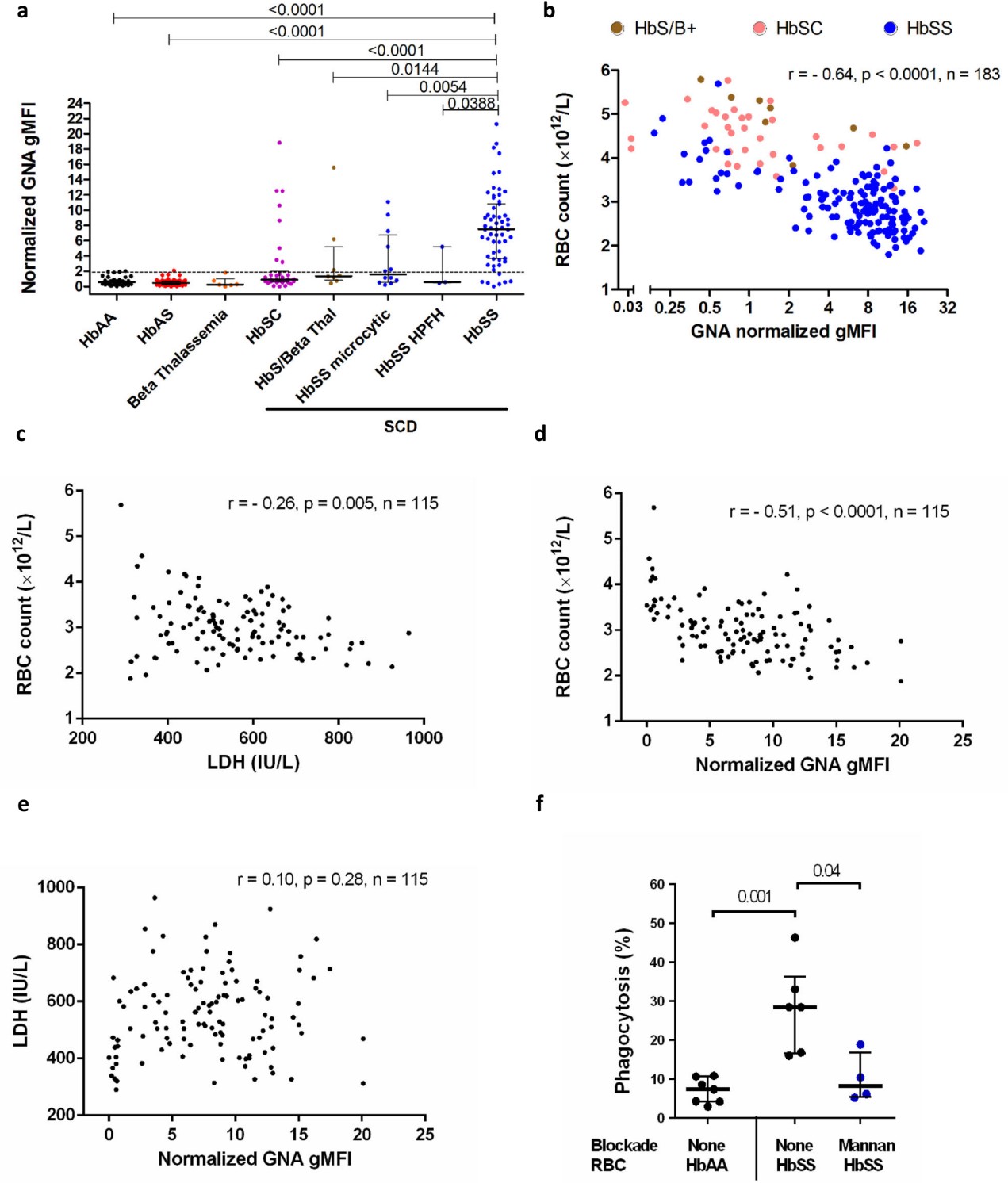

patches observed compared to unoxidized HbSS RBCs (Fig. 3b). Artifactual GNA lectin-binding resulting from permeabilization of the cell by oxidative damage was ruled out (Supplementary Fig. 5c), as was potential intracellular O-GlcNAc binding by GNA lectin (Supplementary Fig. 5d). To identify the cognate receptor on macrophages, we measured the binding of a panel of recombinant mammalian C-type lectin fusion proteins with different glycan specificities to oxidized RBCs. This survey implicated the mannose receptor[17,18] (MR, CD206), in particular the mannose recognizing carbohydrate recognition domain (MR-CRD)

(Fig. 3c). We also observed that in cultures of oxidized RBCs with human monocyte-derived macrophages (HMDM) in vitro, uptake was restricted to MR positive cells (Fig. 3d). Furthermore, siRNA knockdown of MR in macrophages (Supplementary Fig. 6c) specifically reduced phagocytosis (Fig. 3e, Supplementary Fig. 6d); and phagocytosis was also inhibited by the competing glycans mannan or chitin, each known to block MR-CRD,[19] but not inhibited by the control glycan laminarin (Fig. 3f, Supplementary Fig. 6e). Finally, MR-CRD-blocking antibody also inhibited phagocytosis of both oxidized healthy and native HbSS

**Fig. 2 Mannose expression correlates with clinical anemia in sickle cell disease. a** Normalized geometric mean fluorescence (gMFI) of *Galanthus nivalis* Agglutinin (GNA) lectin staining of RBCs from peripheral blood samples, comparing RBC from patients with sub-types of sickle cell disease (milder phenotypes: HbSC (compound heterozygotes for hemoglobin S with C) ($n = 34$), HbS/beta-thal ($n = 8$)(compound heterozygotes for hemoglobin S with β-thalassemia), HbSS microcytic ($n = 12$)(compound heterozygotes for hemoglobin S homozygosity with α-thalassemia) and HbSS HPFH ($n = 3$)) (compound heterozygotes for hemoglobin S homozygosity with hereditary persistence of fetal hemoglobin) versus healthy donors ($n = 45$), sickle cell trait ($n = 57$)(HbAS) and β-thalassemia ($n = 6$). The dotted line indicates the 90th centile of GNA lectin binding within healthy samples. Data are shown as median $+/-$ IQR, 2 tailed Mann–Whitney, p values shown, numerous (>20) experiments. 2 tailed p values relative to HbSS: HbAA, $p = 3.9 \times 10^{-14}$, HbAS, $p = 3.9 \times 10^{-16}$, HbSC, $p = 1.8 \times 10^{-6}$. Each data point derived from an independent RBC donor. No adjustments made for multiple comparisons. **b** Plot of RBC count against normalized GNA gMFI for sickle cell disease: HbS/B + and HbSC indicates compound heterozygosity for HbS with β-thalassemia and HbC respectively; Spearman's rank correlation, $p = 7.0 \times 10^{-23}$, numerous (>20) experiments. Plots of: (**c**) RBC count versus serum lactate dehydrogenase (LDH), Spearman's rank, d) RBC count vs GNA binding, Spearman's rank, ($p = 5.2 \times 10^{-9}$) **e** LDH vs. GNA lectin binding; HbSS RBCs for which corresponding serum LDH values were available; Spearman's rank correlation, numerous (>20) experiments. **f** Percentage phagocytosis of HbAA and HbSS RBCs by human monocyte-derived macrophages analyzed by microscopy. Mannan inhibition as shown. $n = 4$–7 independent RBC donors over two independent experiments. Data are shown as median $+/-$ IQR, 2 tailed Mann–Whitney.

RBC (Fig. 3g, h), which was also sensitive to mannan and chitin (Fig. 3i). Taken together, these results demonstrate that phagocytosis of mannose-displaying RBC is dependent on MR, although the involvement of other receptors cannot be excluded.

**GNA lectin binding proteins comigrate with spectrin containing complexes**. The above data show that high mannose sugars occur in the glycomes of both HbAA and HbSS RBC and that these sugars can be detected on the surface of HbSS RBC and oxidized HbAA RBC by GNA lectin binding. To identify the proteins carrying the high mannose sugars, extracts of HbAA and HbSS RBC membranes were analyzed by western blots probed with GNA lectin. A GNA-binding doublet around 260 kDa was identified in both HbAA and HbSS RBC (Fig. 4a), which is similar in molecular weight to the abundant membrane skeleton proteins α-spectrin and β-spectrin. Blots of HbSS ghosts showed additional GNA lectin-binding bands at ~160 kDa, ~100 kDa, ~70 kDa, and ~50 kDa, which were not seen in fresh HbAA and HbAS RBC (Fig. 4a-b). When HbAA RBCs were stored for six weeks, to allow oxidative damage to membrane skeletal proteins,[20] lower molecular weight Endo-H (N-glycan specific glycosidase) sensitive GNA lectin-binding bands corresponding in size to fragments seen in HbSS cells were seen on western blotting, and GNA lectin precipitation enriched these fragments (Supplementary Fig. 7a, b). The intensity of the 100 kDa fragment was noted to correlate positively with RBC surface GNA lectin binding assessed by flow cytometry (Fig. 4b, Supplementary Fig. 7c), suggesting a role in the surface exposure of high mannose glycans. The specificity of the GNA lectin binding was confirmed by treating RBC ghosts with N-glycan specific glycosidases (PNGase F and Endo-H) prior to western blotting, which abolished GNA-binding to all of the above bands (Fig. 4c). We next attempted to determine whether treatment of RBC ghosts with N-glycanases reduced the sizes of GNA-binding bands. A molecular weight change was not reproducibly observable for the high molecular weight doublet around 260 kDa, although the large sizes of these proteins made minor shifts difficult to observe. However, a ~70 kDa band from HbSS ghosts did show an appropriate reduction in molecular weight, consistent with cleavage of N-glycans after treatment of RBC ghosts with PNGase F and Endo-H (Fig. 4d). Western blotting with antibodies to β-spectrin indicated the band contained an epitope derived from spectrin.

To investigate the association of spectrin with high mannose glycans species further, we carried out mass spectrometric analysis of tryptic peptides from the 260 kDa GNA-binding doublet from both HbAA and HbSS RBC ghosts and found that both bands contained large quantities of α-spectrin and β-spectrin (Supplementary Table 2). As has been previously reported in RBC proteomic experiments,[21] other abundant RBC proteins of lower molecular weight were also identified, including integral membrane glycoproteins such as Band 3 and Glut-1 (Supplementary Table 2). However, despite extensive mass spectrometric analyses from the 260 kDa doublet, no conventionally glycosylated peptides were identified.

Further evidence indicating covalent linkages between high mannose glycan containing glycoproteins/glycopeptides and spectrin derived peptides came from western blots of membrane extracts from HbSS RBC or HbAA ghosts treated with trypsin, which exhibited anti-spectrin binding lower molecular weight bands comigrating with GNA lectin signals (Supplementary Fig. 7c, d), particularly marked for the 50 kDa GNA-binding band and α-spectrin antibodies. GNA lectin precipitation of extracts from HbAA ghosts followed by western blotting with antibodies to spectrin detected a ~260 kDa protein (Fig. 4e). Finally, super-resolution imaging of permeabilized HbAA and HbSS RBC demonstrated GNA lectin co-localizing with spectrin in discrete patches scattered in the spectrin membrane skeletal network (Fig. 4f). Taken together these data support the hypothesis that spectrin-containing complexes in HbSS and oxidized HbAA RBCs are N-glycosylated with high mannose glycans, although it was not possible to detect specific N-glycosylated peptides through conventional glycoproteomic approaches.

**GNA lectin binds to low molecular weight complexes that include spectrin, exhibit protease resistance, and derive from higher molecular weight aggregates**. In order to generate smaller fragments of high mannose-bearing fragments that would be more amenable to characterization, HbSS ghosts were incubated with serial dilutions of trypsin, spectrin was purified from them, and then probed with GNA in western blots. This showed that high concentrations of trypsin, sufficient to digest the 260 kDa and 160 kDa GNA lectin binding proteins, failed to degrade the ~50 kDa, ~70 kDa, and ~100 kDa GNA lectin binding gel bands (Fig. 4g and Supplementary Fig. 7e)). Indeed, the intensities of these bands, particularly that at ~100 kDa, increased with higher trypsin concentrations (Fig. 4g). Prolonged, high concentration trypsin digestion eventually degraded the ~100 kDa fragment, and, to some extent, the ~70 kDa GNA lectin binding bands, with the concurrent appearance of a new GNA lectin binding band around 40 kDa, which we term F40 (Supplementary Fig. 7f). This same pattern of loss of the ~100 kDa and ~70 kDa GNA lectin binding fragments with the concurrent appearance of F40 was also observed when HbSS erythrocytes were stored over five weeks (Supplementary Fig. 7g). Hence, protease digestion results in the formation of a 40 kDa protease-resistant fragment that binds GNA lectin and therefore carries high mannose glycans.

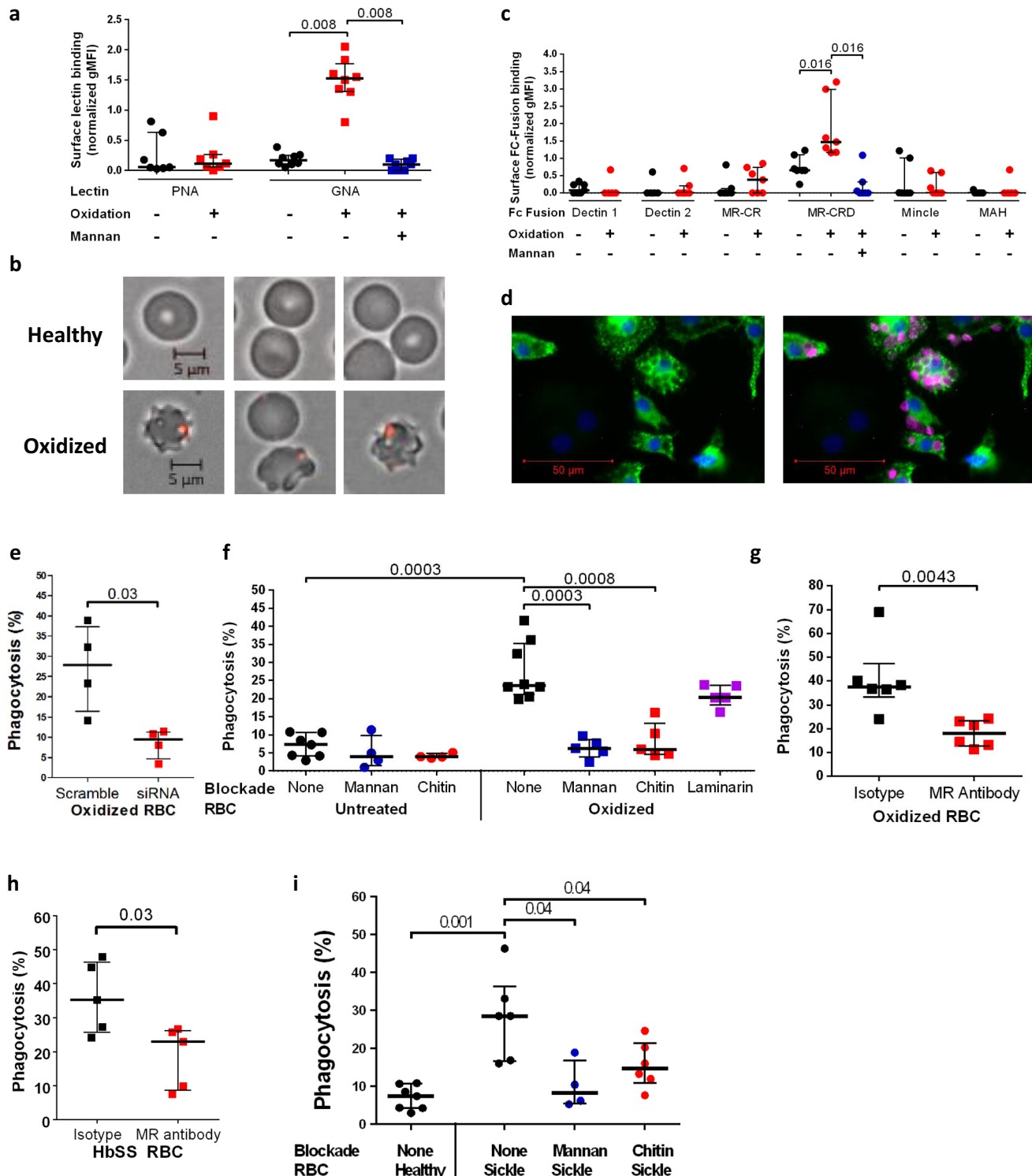

The F40 fragment was concentrated through sequential 100 kDa and 10 kDa cut-off concentrators (Supplementary Fig. 8a, b), and the resulting band cut out from a gel for glycoproteomic analysis. Proteomic analysis of F40 identified peptides from α-spectrin, particularly the N-terminal 370 amino acids (Fig. 4h, Supplementary Table 3). PNGase-F treatment of the purified F40 fragment released N-glycans consisting mainly of high mannoses ($Man_6$-$Man_9$) and complex structures (Supplementary Fig. 8c). However, once again, no

specific glycopeptides could be identified. We postulated that the reason for this, and the relatively low identified protein sequence coverage, could be unconventional peptide structures arising from oxidized and glycated aggregates. Indeed, mass spectrometry confirmed the presence of lysine glycation in α-spectrin peptides from F40 at amino acids K59, K270, and K281. Furthermore, although GNA lectin binding to F40 was Endo-H sensitive, the enzyme required denaturing conditions to be effective (Fig. 4i), consistent with the possibility of protein

**Fig. 3 Display of membrane skeleton-associated mannose patches is induced by oxidative stress and recognized by the mannose receptor on macrophages. a** Peanut agglutinin (PNA) and *Galanthus nivalis* Agglutinin (GNA) lectin binding to normal hemoglobin (HbAA) RBCs with or without oxidation. Mannan blockade for GNA lectin binding shown in blue. 2 tailed Wilcoxon, paired data, PNA, n = 7; GNA, n = 8 biologically independent RBC donors over two independent experiments. **b** Immunofluorescence microscopy of GNA lectin/streptavidin (red) staining of healthy HbAA RBCs (above) and after the oxidative insult (below). **c** Normalized geometric mean fluorescence (gMFI) for binding analyzed by flow cytometry of murine Fc fusions with C-type lectins or sub-domains applied to oxidized versus undamaged RBCs. Mannan blockade of mannose receptor-carbohydrate recognition domain (MR-CRD) binding is shown in blue. MAH, macrophage antigen H. MR-CR, mannose receptor cysteine-rich domain. 2 tailed Wilcoxon, paired data, n = 7 biologically independent RBC donors over two independent experiments. **d** Immunofluorescence microscopy image of human monocyte-derived macrophages (HMDM) stained with DAPI (blue) and for mannose receptor (green) after incubation with oxidized HbAA RBCs, shown in magenta. **e** Percentage phagocytosis of oxidized RBCs by HMDM treated with human MR specific or scrambled siRNA. 2 tailed Mann–Whitney, n = 4 biologically independent RBC donors over two independent experiments. **f** Percentage phagocytosis of healthy or oxidized HbAA RBCs by HMDM with or without pre-blocking by mannan, chitin, or laminarin, 2 tailed Mann–Whitney, n = 4–8 biologically independent RBC donors over two independent experiments. No adjustments made for multiple comparisons. **g** As **f** but oxidized RBCs are blocked by MR-CRD blocking antibody 15.2 as indicated. 2 tailed Mann–Whitney, n = 6 biologically independent RBC donors over two independent experiments. **h** Percentage phagocytosis of sickle cell homozygote (HbSS) RBCs with or without pre-blocking by MR-CRD blocking antibody 15.2. 2 tailed Mann–Whitney, n = 5 biologically independent RBC donors over three independent experiments. **i** HbAA unblocked and HbSS unblocked or mannan and chitin blockade phagocytosis experiments as shown. 2 tailed Mann–Whitney, n = 4-7 biologically independent RBC donors. No adjustments made for multiple comparisons. All data are shown as median +/− IQR.

aggregates. In addition, the N-terminal α-spectrin antibody, B12, failed to bind to the full-length F40 band, but bound to smaller fragments after treatment with a combination of proteases, indicating a cryptic epitope (Supplementary Fig. 8d, e, f). Finally, when visualized in 3D-SIM, some HbSS cells show large aggregates of intracellular spectrin, which correspond to dense GNA lectin surface staining (Fig. 4j). Overall, our data demonstrate the existence of high mannose glycans in HbSS RBC extracts and suggest that the main GNA lectin-binding molecules are spectrin-containing glycoprotein complexes with atypical structures, including glycated forms, that make analysis by conventional glycoproteomics challenging.

**Infection of RBCs with *P. falciparum* causes exposure of high mannose N-glycans, especially those from donors with sickle cell trait.** As infection of RBC with malarial parasites is associated with oxidative stress,[22] we investigated whether exposure to high mannose N-glycans might be important in protection against infection with *P. falciparum*, particularly in the context of sickle cell trait. First, we determined the sensitivity of HbAS RBCs to given oxidative stress and found they bound more GNA lectin than HbAA RBC (Fig. 5a). Interestingly, the proportion of HbS correlated well with the degree of oxidative stress (Fig. 5b). These data suggested that exposure to high mannose N-glycans might contribute to the resistance of individuals with sickle cell trait to severe malaria, by enhancing the clearance of infected cells. We, therefore, infected HbAA and HbAS RBC with *P. falciparum* and assessed high mannose N-glycan exposure as the infection progressed through ring, trophozoite, and schizont stages. HbAA RBCs containing schizonts, but not trophozoites, expressed significantly higher exposed high mannose N-glycans as indicated by increased GNA lectin binding (Fig. 5c). Importantly, HbAS RBCs containing schizonts expressed even higher levels of exposed high mannose N-glycans, and this increased expression extended into the trophozoite stages (Fig. 5c). *P. falciparum* infected RBCs cytoadhere to vascular endothelium, to avoid phagocytosis by hepatosplenic macrophages,[23,24] and this adhesion is mediated by the expression of PfEMP1 on late-stage infected RBC.[25] Reduced display of PfEMP1 on the surface of HbAS-infected RBCs is a potential mechanism of protection against malaria in sickle cell trait,[26] and PfEMP1 levels show considerable variation in different HbAS donors.[27] We, therefore, determined PfEMP1 expression in addition to mannose display and noted a marked inverse correlation (Fig. 5d).

## Discussion

This work has identified a receptor-ligand pair mediating RBC clearance that underlies both the extravascular hemolysis of sickle cell disease and clearance of *P. falciparum* infected RBCs. Immunologically, the latter can be regarded as protective immunity arising from the recognition of altered self, with the mannose receptor recognizing a pattern common to both diseased and infected cells. The display of high mannose N-glycans on membrane proteins could also be regarded as a pathogen-associated/damage-associated molecular pattern (P/DAMP). The mannose receptor is expressed in human spleens by Lyve-1 + cells lining venous sinuses, where they form a physical barrier for blood cells to exit the red pulp and so are ideally located to perform a filtering function.[28] In infection with *P. falciparum*, the parasite evades passage through the spleen by expressing adhesive proteins, notably PfEMP1, on the surface of infected RBCs, so that they adhere to endothelial cells in the systemic circulation. The inverse correlation between mannose exposure and surface PfEMP1 implies similar processes involving oxidation-induced membrane skeletal rearrangements underlie both phenomena.[29,30] Reduced PfEMP1 in infected HbAS RBCs will lead to a failure of cytoadherence, and the exposed high mannose N-glycans on circulating infected RBC will induce clearance by hepatosplenic phagocytes. Together with the spleen's role in processing high mannose N-glycan bearing RBCs in sickle cell disease, these data are consistent with the spleen being the primary organ in removing RBCs infected with malaria exposing high mannose N-glycans. Our work also potentially sheds light on the reasons why those with sickle cell disease are so susceptible to infections with encapsulated bacteria, especially *Streptococcus pneumoniae*, which is the commonest cause of death in children.[1] Capsular polysaccharide from pneumococcus is known to bind the carbohydrate-binding domains of the mannose receptor.[31] It, therefore, seems likely that the high mannose N-glycans on the surfaces of sickle cells would compete with bacteria for uptake by the mannose receptor.

Our work has also identified a phenomenon whereby complexes of membrane skeletal proteins, and fragments derived from them, are associated with high mannose N-glycans, which act as an eat-me signal. There are still only two accepted eat-me signals in higher eukaryotes, phosphatidylserine, and calreticulin.[32] This work, therefore, adds a third ligand to perform this role. The expression of ligands for uptake attached to the membrane skeleton would allow receptors on phagocytic cells to bind to molecules with high tensile strength, which may be important

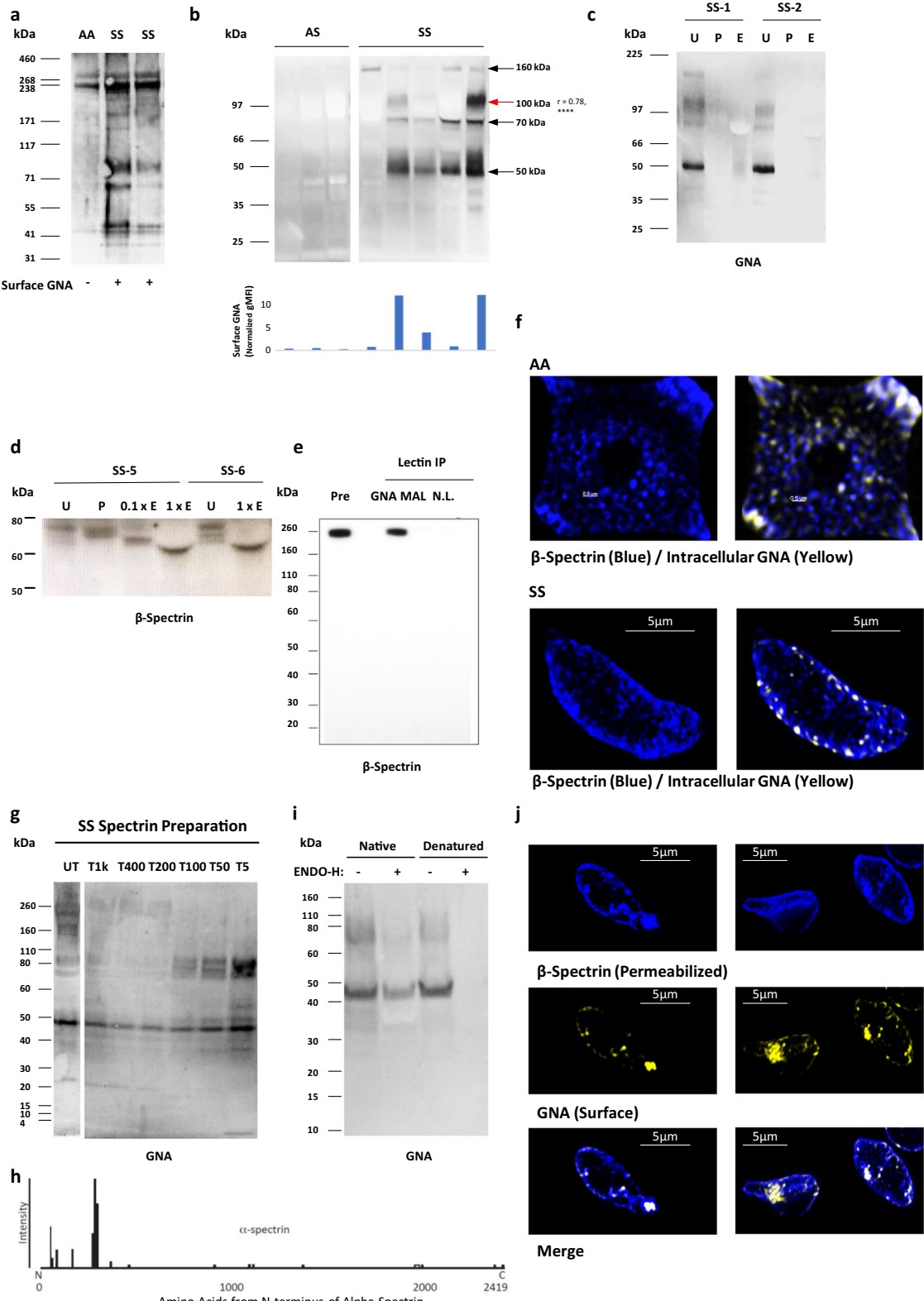

for capturing cells as they transit through the spleen under conditions of high shear stress.

Protein aggregates are thought to form after attacks by free radicals, reactive oxygen and nitrogen species, and glycation. These result in a wide variety of amino acid adducts,[33] some of which mediate the cross-links thought to underlie the formation of aggregates. Complexes of damaged proteins, including spectrin, secondary to oxidative damage associated with hemolysis have long been recognized in RBCs,[34–37] partly arising from interaction with free radicals generated by denatured hemoglobin species (hemichromes), including in sickle cell disease.[38] Oxidative degradation has been shown to be one of the main causes of

**Fig. 4 High mannose decoration of spectrin containing fragments in sickle cell disease. a** *Galanthus nivalis* Agglutinin (GNA) lectin western blot from healthy (HbAA) and homozygote sickle hemoglobin (HbSS) ghosts. **b** Above are shown further GNA lectin western blots from HbAA and HBSS ghosts. The histogram below the blot shows the flow cytometrically measured surface GNA lectin staining values of the RBCs used to make the ghosts, with each bar corresponding to the cells used to make the western lane above. The *r*-value to the right of the 100 kDa size label is Spearman's rank correlation coefficient between GNA lectin staining values and band intensities, both classified ordinally as high, medium, or low (post hoc analysis $p = 1.3 \times 10^{-6}$, $n = 27$ measurements from 22 individuals, no statistical adjustment made for other bands). **c** GNA lectin blot from HbSS ghosts: untreated (U), treated by PNGase (P) or Endo-H (E). **d** High exposure β-spectrin blot showing PNGase and partial/full (0.1×/1×) Endo-H digestion of two HbSS ghosts. **e** Lectin precipitation of healthy ghosts with GNA or *Maackia amurensis* Lectin II (MAL-II) lectins. No lectin control is also shown. Immunoblot with the β-spectrin specific antibody. **f** Super-resolution microscopy image of spectrin membrane skeleton (blue) from healthy (AA) and sickle cells (SS). Yellow clusters of GNA staining are overlaid. 3D image is sliced to reveal a single sheet of membrane skeleton network. **g** GNA lectin blot of spectrin released from HbSS ghosts after digestion with trypsin for one hour. Untreated (UT), Tx indicates the dilution factor of trypsin relative to spectrin material. **h** The amino acid sequence of α-spectrin showing peptide coverage (presence or absence of bar above the line) and intensity (proportional to the height of the bar) from proteomic analysis of F40 following chymotrypsin treatment. **i** GNA lectin blots showing Endo-H treatment of the 10 kDa concentrate from **e** under native or denaturing conditions (urea/SDS/2-mercaptoethanol) for 24 h. **j** 3D SIM super-resolution microscopy of surface GNA lectin binding and internal β-spectrin in HbSS RBC. Cells are first stained with GNA lectin (yellow), then permeabilized, and stained with anti-spectrin antibody (blue) (**k**).

membrane skeletal protein alterations occurring in RBCs in storage, with proteolytic cleavage having a secondary role.[20] Interestingly, spectrin-containing species were detected either as low molecular weight fragments covering the N-terminus, as found in our proteomic analysis, or as high molecular aggregates.[20] Oxidation during the storage period of RBCs has also been shown to inactivate glyceraldehyde 3-phosphate dehydrogenase, an important enzyme for ATP synthesis.[20,39] In turn, this leads to dissociation of spectrin from the phosphatidylserine molecules of the RBCs membrane, in an ATP dependent mechanism, resulting in increased spectrin-glycation products.[40]

Therefore, either by directly acting on the cytoskeletal proteins, or indirectly through ATP-dependent mechanisms, oxidative damage of RBCs is a mechanism that induces alterations in membrane protein organization leading to aggregation of membrane glycoproteins. This is in accordance with a recent report demonstrating that oxidative stress results in cluster-like structures on the membrane of RBCs as a result of possible reorganization and aggregation.[41] RBCs contain various glycoproteins such as band 3, glycophorin, GLUT1, CD44, and CD47.[42] Therefore, the enhanced phagocytosis we describe could potentially be driven by the aggregation of RBC membrane glycoproteins increasing the local concentration of high mannose N-glycans, thus favoring their recognition by the mannose receptor. This is in accordance with previous reports showing that the binding affinity of the mannose receptor increases with the density of mannose-containing glycoproteins.[43] Taken together, we suggest that oxidative stress in RBCs induces glycoprotein reorganization and aggregation, resulting in increased high mannose glycan bearing densities that are recognized by the mannose receptor.

In summary, we describe a mechanism whereby oxidatively-damaged, membrane protein complexes display high mannose N-glycans, which act as eat me signals important in the hemolysis of sickle cell disease and resistance against severe malaria. It, therefore, represents a unified mechanism to explain both advantageous and deleterious consequences of the sickle mutation.

## Methods

**Donors.** Ethical approval was obtained for the study (North of Scotland REC Number 11/NS/0026) and any blood obtained using this approval was collected after informed consent. Further samples, including donors with sickle cell trait, were obtained from the NHS Grampian Biorepository scheme (application number TR000142), which provided anonymized samples from routinely collected clinical samples once all requested tests had been completed and tubes were ready for disposal. No extra blood samples were taken for research purposes in this scheme. Anonymized samples were also obtained from patients with sickle cell disease from King's College Hospital, where samples were used from routinely collected clinical

samples once all requested tests had been completed. No extra blood samples were taken for research purposes. Patients were informed that discarded, anonymized samples might be used for research by displaying a written notice. Hemoglobin contents were assessed by HPLC (Tosoh G7). All samples from patients were collected into EDTA tubes (Becton-Dickinson).

**RBC isolation.** Blood was collected into acid citrate dextrose solution tubes (ACD; 455055, Grenier) and RBC isolated by sodium metrizoate density gradient centrifugation (1.077 g/ml, Lymphoprep; 1114547 Axis-Shield). Packed RBC was diluted with an equal volume of Dulbecco's modified Eagle's medium (DMEM; 4.5 g/L glucose, L-glutamine; 41965, Gibco), stored in ACD (9 ml RBC/DMEM per ACD tube) at 4 °C and used within 3 days unless otherwise stated.

**Lectins.** Biotinylated lectins were all purchased from Vector Laboratories. They include: *Galanthus nivalis* Agglutinin (GNA, B-1245, 4 μg/ml), *Narcissus pseudo-narcissus* Lectin (NPL, B-1375, 4 μg/ml), *Griffonia simplicifolia* Lectin II (G.Simp, B-1215, 4 μg/ml), *Solanum tuberosum* Lectin (STL, B-1165, 20 ng/ml), *Aleuria aurantia* Lectin (AAL, B-1395, 33 ng/ml), *Maackia amurensis* Lectin II (MAL II, B-1265, 67 ng/ml), *Sophora japonica* Lectin (Vector Laboratories, no longer available, 1 μg/ml). FITC conjugated Peanut agglutinin was purchased from Sigma-Aldrich (PNA, L7381-2MG, 2 μg/ml).

**Flow cytometry.** Whole blood flow cytometry assays were used for Figs. 1a, 2, Supplementary Figs. 1a, 2, 3b, d, e, and 8a. Whole blood was washed first in phosphate-buffered saline (PBS). Purified RBCs were used for the other flow cytometry experiments. RBC gating was applied by forward and side scatter gating of both whole blood and purified RBC flow cytometry experiments (Supplementary Fig. 9a). Staining with anti-glycophorin A (GPA) confirmed gates contained >99% RBC (data not shown).

For lectin flow cytometry, approximately $5 \times 10^6$ RBC were washed three times in PBS and incubated for 15 min at room temperature in calcium buffer (10 mM HEPES, 150 mM NaCl₂, 2.5 mM CaCl₂, pH 7.4) containing 10% Carbo-Free Blocking Solution (SP5040, Vector Laboratories) for whole blood flow cytometry or just buffer alone for purified RBC flow cytometry. Biotinylated lectin staining was carried out at room temperature in the same buffer as the initial blocking step. PNA-FITC and other antibody staining were carried out in PBS throughout, without contact with calcium buffer or Carbo-Free Blocking Solution. Lectin and antibody staining was carried out for 30 min at room temperature, protected from light. Annexin V staining was carried out according to the manufacturer's instructions (640945, Biolegend). For biotinylated lectin staining, cells were then washed and incubated with streptavidin PE-Cy7 (0.27 μg/ml; 25431782, eBioscience) or PE (0.67 μg/ml; 554061, BD Pharmingen) for 30 min at room temperature. Humanized Fc fusions of murine C-type lectins[17,44] (5 μg/ml) were incubated with RBC for 30 min at room temperature in calcium buffer, then detected by Alexa Fluor 647 goat anti-human secondary antibody (2 μg/ml; 109-605-098, Jackson ImmunoResearch Laboratories). In tests of their specificity for binding RBC, lectin or Fc fusion proteins were first incubated with mannan (5 mg/ml, unless otherwise stated) for 15 min at room temperature. Biotinylated BRIC 132 and BRIC 163 (10 μg/ml; 9458B and 9410B, International Blood Group Reference Laboratory) and anti-O-GlcNAc (1 μg/ml, RL2; 59624, Santa Cruz) binding to RBC were performed in PBS for 30 min at room temperature, then incubation with streptavidin secondary (for BRIC 132/163) or anti-mouse PE secondary (for anti-O-GlcNAc) for a further 30 min.

Prior to intracellular staining, RBC was fixed with glutaraldehyde (0.05%, 10 min, room temperature), permeabilized with Triton X-100 (0.1% in PBS) for 5 min at room temperature, and then washed in PBS. Cells were washed before cytometric analysis. Data were acquired on a FACSCalibur (BD) and analyzed using FlowJo v10.0 (Treestar) software. Normalized geometric mean fluorescences (gMFI) were calculated by subtracting the gMFI of secondary antibody/

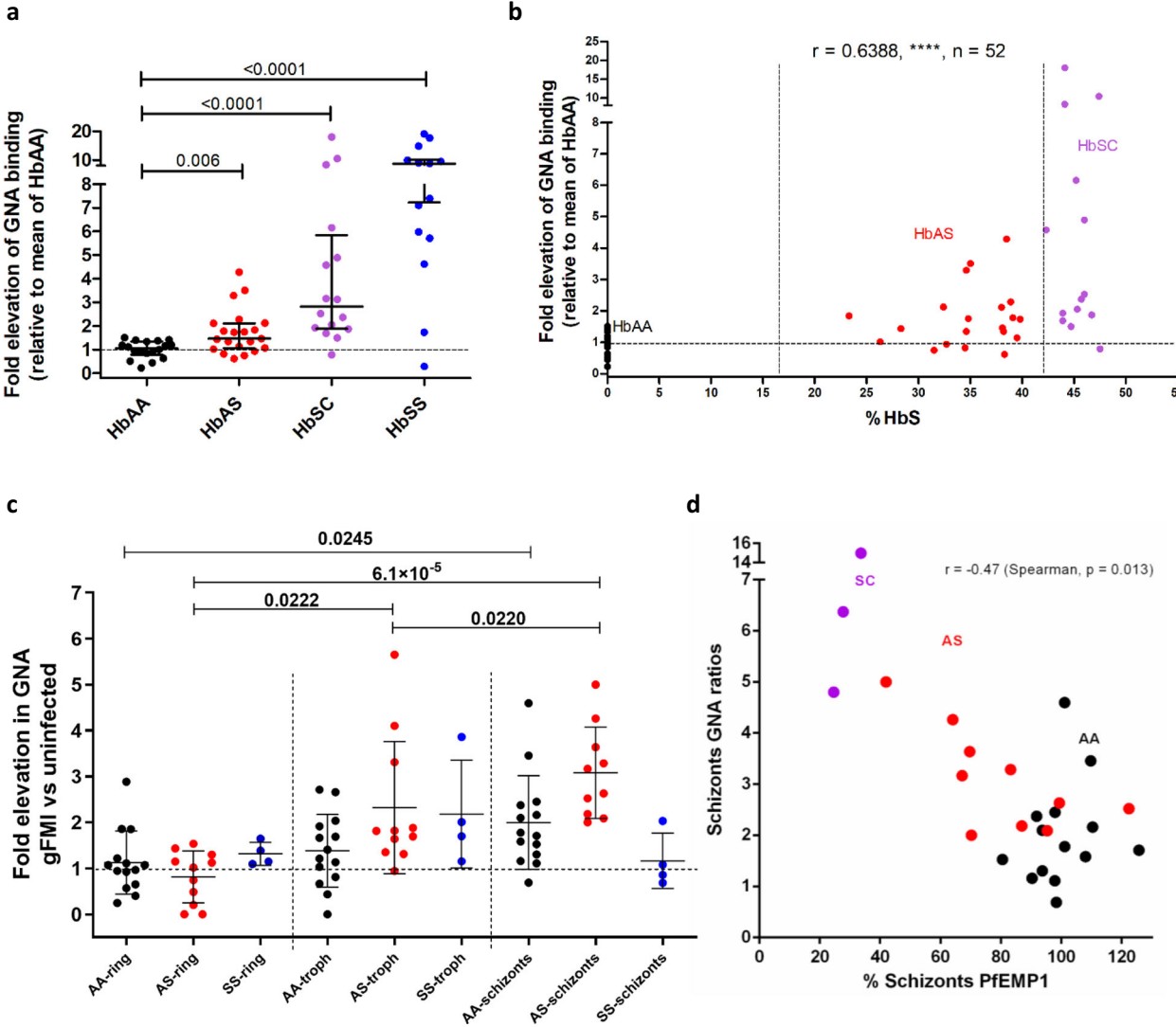

**Fig. 5 High levels of mannose are displayed by sickle cell trait RBCs in response to both oxidative stress and infection with *P. falciparum*. a** *Galanthus nivalis* Agglutinin (GNA) lectin binding to healthy (HbAA) ($n = 18$) versus sickle cell trait (HbAS) ($n = 21$), compound heterozygotes for hemoglobins S and C (HbSC) ($n = 16$), homozygotes for hemoglobin S (HbSS) ($n = 14$) RBCs in response to oxidative stress, expressed as a ratio to mean of oxidized HbAA samples. Data are shown as median $+/-$ IQR, 2 tailed Mann–Whitney. *p* values relative to HbAA: HbSC, $p = 8.6 \times 10^{-6}$, HbSS, $p = 3.5 \times 10^{-5}$. Each data point indicates an independent RBC donor over more than five independent experiments. **b** HbS percentage is plotted against elevation of GNA lectin binding in response to oxidative stress as in **a**. (****$p = 3.5 \times 10^{-7}$). HbAA, HbAS, and HbSC donor samples are shown as indicated. Spearman's rank statistics shown. c) GNA binding to HbAA, HbAS, and HbSS RBCs in response to infection with *P. falciparum*. Values for rings, trophozoites, and schizonts are expressed as ratios relative to the uninfected gate. Linked one-way ANOVA analysis for HbAA ($p = 0.027$), HbAS ($p = 9.5 \times 10^5$), HbSS (n.s. $p = 0.192$); Tukey's multiple comparisons within each genotype. $n = 27$ independent RBC donors over five independent experiments. Data are shown as mean $+/-$ SD. **d** Plot of relative GNA lectin binding for RBCs infected by schizonts against relative PfEMP1 expression. Hemoglobin phenotypes of donors as indicated, Spearman's rank.

streptavidin-only paired controls. For PNA-FITC and Annexin V analysis, unstained controls were used for gMFI normalization.

**RBC ghost preparation**. Washed RBCs were subjected to ice-cold hypotonic lysis in 20 mM Tris, pH 7.6, with protease inhibitor (05056489001, Roche).[45] Lysates were washed three times in hypotonic lysis buffer ($37,000 \times g$, 4 °C, 30 min) before resuspension in minimal hypotonic lysis buffer. Protein concentrations were determined by protein BCA assay (23227, Pierce). No trypsinization was performed before any glycan analysis.

**Glycomic mass spectrometry (MS)**. N-linked glycan analysis from RBC ghosts was performed according to Jang-Lee et al.[46]. MS and MS/MS data from the permethylated purified glycan fractions were acquired on a 4800 MALDI-TOF/TOF mass spectrometer (Applied Biosystems). Data were processed using Data Explorer 4.9 Software (Applied Biosystems). The processed spectra were subjected to manual assignment and annotation with the aid of a glycobioinformatics tool,

GlycoWorkBench.[47] Proposed assignments for the selected peaks were based on [12]C isotopic composition together with knowledge of the biosynthetic pathways, and structures were confirmed by data obtained from MS/MS experiments.

**Proteomics**. RBC ghost membranes were subjected to SDS-PAGE (10% Bis-Tris gel with MOPS running buffer). SDS-gel bands were excised, sliced into small pieces, and destained with 200 µl of 1:1 v:v acetonitrile:ammonium bicarbonate (50 mM, pH 8.4; AMBIC). Destained gel pieces were then reduced by treatment with 10 mM DTT in AMBIC at 56 °C for 30 min, carboxymethylated in 55 mM iodoacetic acid in AMBIC in the dark at room temperature, and then subjected to overnight sequencing grade modified trypsin (Promega V5111) digestion in AMBIC at 37 °C. After enzyme inactivation (100 °C water bath, 3 min), the digested peptides were extracted twice from the gel pieces by incubating sequentially (15 min with vortexing) with 0.1% trifluoroacetic acid and 100% acetonitrile. Finally, the volume was reduced with a SpeedVac. Eluted peptides were analyzed by LC–MS using a NanoAcquity UPLC™ system coupled to a Synapt™ G2-S mass

spectrometer (Waters MS Technologies, Manchester, UK) in positive ion mode. Five microliters of the sample was injected onto the analytical column (Waters, HSS T3, 75 μm × 150 mm, 1.8 μm). For the 260kD band, peptides were eluted according to the following linear gradient program (A: 0.1% v/v formic acid in the water, B: 0.1% v/v formic acid in acetonitrile): 0–90 min, 3–50% of B. MS data were acquired on the Synapt G2-S using a data-dependent acquisition program, calibrated using Leu-Enkephalin peptide standard. The top 20 components were selected for MS/MS acquisition. The F40 band was analyzed with MS$^e$ with 10–40% of B. MS$^e$ is a feature of the Waters Q-TOF mass spectrometer acquisition control allowing the alternating generation of total fragment ion data from the ion beam, switching between high and low collision energies, without the prior individual mass selection of DDA. Identification of the eluted peptides was performed using ProteinLynx Global SERVER™ v3.03 (Waters) using a human porcine trypsin database (Uniprot 1.0). The following were set as workflow parameters on PLGS: fixed carboxymethyl modification for cysteine, variable deamidation, and oxidation modifications for glutamine and methionine respectively.

**Human monocyte-derived macrophage (HMDM) preparation and culture**. Mononuclear cells were isolated by density centrifugation from whole blood and seeded at $10^6$ cells/ml in Roswell Park Memorial Institute medium (RPMI (21875-034, Gibco)), 100 U/ml penicillin, 100 μg/ml streptomycin, 292 μg/ml L-glutamine (10378-016, Gibco) and 10% heat-inactivated autologous serum. Cultures were incubated at 37 °C with 5% $CO_2$ for 14–21 days. Cells were washed with RPMI three times prior to use.

**Phagocytosis Assay**. For identification of phagocytosis by microscopy, RBC was stained with Cell Trace Far Red (CTFR) (C34564, Molecular Probes) according to the manufacturer's instructions with minor alterations: CTFR was diluted at 1 in 500 (2 μl/ml) in RPMI with penicillin and streptomycin (RPMI/PS) media and incubated with 20 μl packed RBC for 30 min at 37 °C, after which staining was inhibited by adding 10% FCS (10270-106, Gibco). Stained cells were washed in RPMI/PS prior to counting and addition to macrophages. RBC was added to HMDM at $5 \times 10^7$ cells per well for 3 h, before removal of cells, washing, and fixation with 4% paraformaldehyde (Supplementary Fig. 6a). RBC bound, but not ingested, by HMDM were then stained with anti-glycophorin-FITC (HIR2 antibody; 306610, Biolegend). Cells were imaged using an immunofluorescence microscope (Zeiss AxioObserver Z1). Phagocytic macrophages were defined as containing at least one CTFR-positive GPA-FITC-negative RBC (determined by bright field). Three examples of oxidized RBC phagocytosis are shown in Supplementary Fig. 6b, marked P. RBC-binding macrophages were defined as associated with at least one glycophorin-FITC/CTFR double-positive RBC. Analysis of HMDM phagocytosis included only the small, non-granular subset of macrophages, because of consistent association with phagocytosis and binding of RBC. For quantification of phagocytosis, 200-500 such macrophages were counted per treatment. To test the specificity of HMDM recognition, the polymers mannan (10 mg/ml; M-7504, Sigma-Aldrich), chitin (50 μg/ml; C9752, Sigma-Aldrich) or laminarin (10 mg/ml; L9634, Sigma-Aldrich), or anti-CD206 blocking antibody (10 μg/ml clone 15.2 321102, BioLegend; isotype control mouse IgG1 kappa clone 107.3; 554721, BD Biosciences) were added to cultures 60 min before phagocytosis assays. Coumarin-stained 6 μm Fluoresbrite carboxylate microspheres, of similar size to RBC, were used to assess RBC independent phagocytosis (Supplementary Fig. 6e).

**RBC oxidation and eryptosis**. Purified RBC was incubated for 60 or 30 min, respectively, with 0.2 mM copper sulfate and 5 mM ascorbic acid at 37 °C in DMEM with 4.5 g/L glucose. Cells were washed in PBS 3 times prior to use. To induce eryptosis, calcium ionophore (2 μM, A23187, Sigma-Aldrich) was applied at 37 °C in DMEM, with 4.5 g/L glucose, to purified RBC for 3 h.

**Reactive Oxygen Species (ROS) production**. The rate of ROS production was determined by first loading purified oxidized or untreated RBC with oxidation sensitive dye CM-H2DCFDA (10 μM; C6827, Molecular Probes) in PBS and incubating for 60 min in the dark at 37 °C. RBCs were then washed three times, resuspended in DMEM and fluorescence determined immediately by spectrofluorimeter (Fluostar Optima; BMG Labtech) with excitation of 485 nm and emission 530 nm. The rate of ROS formation was calculated for the linear portion of the fluorescence/time curve generated over six hours, which typically lasted for three hours.

**Lectin/Immuno-blotting**. Ghost preparations were mixed in equal volumes with SDS sample buffer containing 8 M urea[45] and heated at 100 °C for 10 min. Ghost protein samples were fractionated by gel electrophoresis using NuPage 4–12% Bis-Tris gel (Invitrogen, NP0312BOX) and transferred by western blotting (30 V, 1 h) to polyvinylidene fluoride membrane (P 0.45 μm; 10600023 Amersham Hybond, GE Healthcare). Blots were probed with biotinylated GNA lectin (40 μg/ml; B1245, Vector Laboratories) and Streptavidin HRP (1:2500 dilution, 3999 S, Cell Signaling) in a calcium-binding buffer (10 mM HEPES, 150 mM $NaCl_2$, 2.5 mM $CaCl_2$, pH 7.4) containing 1× Carbo-Free Blocking Solution (Vector Laboratories, SP-5040) and protease inhibitor cocktail (11836145001, Roche) before development in

Amersham ECL Select substrate (RPN2235, GE Healthcare). 0.1% Tween-20 was added in probing and washing steps. Loading of wells was normalized by protein concentration (~6 μg per sample). Enzymes PNGase F (P0704S, New England Biolabs) and Endo-Hf (P0703L, New England Biolabs) were used according to the manufacturer's instructions to treat RBC ghost samples prior to electrophoresis.

**Lectin precipitation**. RBC ghosts were suspended in equal volumes of calcium-binding buffer containing 2% Triton X-100, pre-cleared with magnetic streptavidin beads (88816, Pierce), and incubated with biotinylated GNA lectin (1 mg/ml; B1245, Vector Laboratories), biotinylated MAL-II lectin (1 mg/ml; B1265, Vector Laboratories) or buffer only overnight at 4 °C. Precipitation with magnetic streptavidin beads was performed in a binding buffer and the beads washed with a binding buffer containing 0.1% Triton X-100. Washed precipitates were denatured at 100 °C for 10 min and supernatants loaded for gel electrophoresis and blotting. Blots were probed with anti-spectrin antibody (1 in 20,000 dilutions; S3396, Sigma-Aldrich) and anti-mouse-HRP secondary antibody (1 in 10,000 dilutions, 5887, Abcam). PBS/0.1% Tween 20 replaced calcium-binding buffer for lectin blotting.

**Spectrin purification**. Spectrin was purified following the method of Ungewickell et al. with slight modifications.[48] Briefly, the ghosts were washed twice and resuspended in 3 volumes of 37 °C pre-warmed sodium phosphate (0.3 mM, pH 7.2) (extraction buffer) and incubated for 20 min at 37 °C. The fragmented ghosts were pelleted by centrifugation at 40,000 × g for 1 h at 2 °C. The supernatant was used as spectrin preparation for analysis.

**Serial trypsin dilution treatment of spectrin**. Spectrin preparations or ghosts were analyzed for protein content by BCA assay. Titrations of trypsin at concentrations as a fraction of sample concentration were applied for one hour or longer if indicated, at 37 °C. No trypsin addition was applied to the untreated sample, which was also incubated for the same duration. Samples were all heat-inactivated at 100 °C for 10 min, after diluting with 8 M urea sample buffer at a 1:1 ratio.

**Chymotrypsin and pepsin digestions**. Chymotrypsin and pepsin were applied at 1:5 dilution in the sample. Combinatorial protease treatment over 48 h was performed with heat inactivation for 100 °C, 10 min at 24 h. During pepsin treatment, the sample was pre-diluted 1:1 with HCl, pH2.0. Acid was neutralized with NaOH after 24 h, prior to the addition of other proteases.

**Isolation of trypsin resistant sickle fragment (TRSF)(F40)**. Approximately 20 ml of HbSS ghosts, having been washed with low cold salt extraction buffer (0.3 M sodium phosphate, pH 7.6), was treated with 1:5 trypsin: sample ratio overnight. Heat inactivation at 100°C was carried out for 10 min. The supernatant was harvested and further centrifuged to remove insoluble products. The clarified supernatant was subsequently concentrated with a 100 kDa cut-off concentrator (Pierce, Thermo Fisher, 88533) and supernatant applied to and concentrated with a 10 kDa cut-off concentrator to approximately 500 μl.

**Mass spectrometry glycomic analysis of TRSF**. Urea containing sample buffer, as above, was applied to TRSF. Coomassie bands corresponding to the 40–44 kD region were cut and analyzed in three segments: 39–40 kD, 41–42 kD, and 43–44 kD.

**Dual-color western**. Amersham western blotting machines were used to detect GNA lectin and anti-spectrin binding using Cy3 and Cy5 conjugated reagents. Data were acquired and analyzed by Amersham's inbuilt software.

**Immunofluorescence microscopy**. Cell surface GNA lectin binding for immunofluorescence was performed as for flow cytometry with minor alterations ($10^7$ cells per test, 8 μg/ml GNA lectin, 1 μg/ml Streptavidin PE. For Fig. 1b and Supplementary Fig.1c and 1d, 8 μg/ml GNA lectin and 1 μg/ml Streptavidin PE were pre-complexed overnight). Intracellular GNA lectin binding followed fixation (0.005% glutaraldehyde/PBS, 10 min, room temperature) and permeabilization (0.1% Triton-X 100/PBS, 15 min, room temperature). Stained cells were pulse centrifuged (≤300 × g) for 30 s (including acceleration), in 24 well, flat bottom tissue culture plates (Greiner). To stain CD206, cells were blocked for 15 min with 1% BSA/PBS at room temperature in the dark and incubated with Alexa-488 conjugated anti-mannose receptor antibody (1.25 μg/ml, Clone 19.2; 53-2069-47, eBiosciences). DAPI (as per manufacturer's instructions; D1306, Thermo Fisher) staining was applied to cells post-fixation/permeabilization for 30 min at room temperature. Cells were washed in PBS and imaged at ×32 magnification using an immunofluorescence microscope (Zeiss AxioObserver Z1). Images were analyzed by Zen (Black and Blue versions, Zeiss).

**siRNA knockdown of mannose receptor**. Human mannose receptor (CD206) siRNA (UACUGUCGCAGGUAUCAUCCA) or a non-targeting siRNA sequence control (4390843, Life Technologies) were transfected into HMDM (RNAiMax,

Life Technologies) ($n = 4$ donors for all siRNA experiments). Knockdown efficiency was established by determining mannose receptor expression by microscopy using CD206-Alexa-488 staining (described above) in the small non-granular macrophage sub-population by merging bright field and mannose receptor fluorescence staining. Knockdown efficiency was typically 65–85% (Supplementary Fig. 6c).

**Confocal microscopy for RBCs.** For spectrin-GNA lectin double-staining experiments, permeabilized RBCs were stained with anti-human spectrin antibody (1 in 50 dilutions; S3396, Sigma Aldrich) concurrently with GNA lectin (8 µg/ml) in calcium buffer. Alexa Fluor 647 anti-mouse antibody (10 µg/ml; A31571, Thermo Fisher) was applied in conjunction with streptavidin PE (1 µg/ml; Beckman Dickinson) following staining of primary reagents. RBCs were gravity sedimented (30 min at room temperature, in the dark) onto poly-L-lysine (Sigma Aldrich) treated 8 well chamber slides (LabTek). Images were acquired by a Zeiss LSM 710 microscope.

**3D-Structured Illumination Microscopy (3DSIM).** GNA lectin/anti-spectrin stained RBCs were gravity sedimented onto poly-L-lysine treated chamber slides (LabTek). 3DSIM images were acquired on an N-SIM (Nikon Instruments, UK) using a 100 × 1.49NA lens and refractive index-matched immersion oil (Nikon Instruments). Samples were imaged using a Nikon Plan Apo TIRF objective (NA 1.49, oil immersion) and an Andor DU-897X-5254 camera using 561 and 640 nm laser lines. Z-step size for Z stacks was set to 0.120 µm, as required by the manufacturer's software. For each focal plane, 15 images (5 phases, 3 angles) were captured with the NIS-Elements software. SIM image processing, reconstruction, and analysis were carried out using the N-SIM module of the NIS-Element Advanced Research software. Images were checked for artifacts using SIMcheck software (http://www.micron.ox.ac.uk/software/SIMCheck.php). Images were reconstructed using NiS Elements software v4.6 (Nikon Instruments, Japan) from a Z stack comprising ≥1 µm of optical sections. In all SIM image reconstructions, the Wiener and Apodization filter parameters were kept constant. Reconstructed SIM images were rendered in 3 dimensions using Imaris (Bitplane). Intracellular GNA lectin and anti-spectrin staining of healthy RBCs were performed following fixation and permeabilization. In order to co-localize surface GNA lectin with intracellular spectrin, HbSS RBCs were stained with GNA lectin/streptavidin-PE staining before, and anti-spectrin/donkey anti-mouse Alexa 647 after, fixation and permeabilization.

***P. falciparum* culture in RBCs of different genotypes and flow cytometry analysis.** *P. falciparum* IT/FCR3 parasites were cultured at 2% hematocrit in supplemented RPMI.[49] Group O + erythrocytes (Scottish National Blood Transfusion Service, Edinburgh, Scotland) and RPMI 1640 (Invitrogen, Paisley, UK) were supplemented with 20 mM glucose (Sigma, Poole, UK), 2 mM glutamine (Invitrogen), 25 mM HEPES (Lonza, Basel, Switzerland), 25 µg/mL gentamicin (Lonza) with 10% pooled normal human serum (Scottish National Blood Transfusion Service) adjusted to pH 7·2–7·4 with NaOH (Sigma). Flasks were gassed with 1% $O_2$/3% $CO_2$/96% $N_2$ and incubated at 37 °C. Mature trophozoite-infected erythrocytes were purified to >90% parasitemia by magnetic separation with a MACS CS column (Miltenyi Biotec, Germany).[50] The purified infected erythrocytes were used to infect RBCs of different genotypes with a starting parasitemia of approximately 0.5%. RBCs were used within 10 days post-bleed, typically 3–5 days. Cultures were gassed with 1% oxygen, 3% carbon dioxide, and 96% nitrogen, then incubated at 37 °C for 48–72 h. After one cycle of invasion and growth, flow cytometry was used to assess parasitemia and parasite maturation,[51] as well as GNA lectin-binding and PfEMP1 antibody staining. This method uses internally controlled flow cytometry analysis to separate uninfected red cells, ring-stage, trophozoite, and schizont stage parasites within the same culture by FACS gating using a pair of DNA-binding and RNA-binding dyes (Supplementary Fig. 9b, c). Hypoxia-induced reduction in parasite invasion and growth was observed in HbAS red cells, as described by Archer et al.[8] However, a range of parasite stages was available within each culture to allow investigation of high mannose exposure as parasites matured through the blood-stage cycle.

For flow cytometry, infected erythrocytes in binding buffer (10 mM HEPES, 150 mM NaCl, 2.5 mM $CaCl_2$, pH 7.4) were stained with Vybrant Violet (Thermo Fisher, V35003, 2.5 µM) and ethidium bromide (Sigma-Aldrich, 46067-50ml-F, 1% in $dH_2O$). Biotinylated GNA lectin staining was performed as described above. Streptavidin APC was used to detect biotinylated GNA lectin. A BD LSRFortessa (BD Biosciences) was used for flow cytometry and compensation between channels was carried out prior to the experiment. Relative GNA lectin binding was calculated for ring, trophozoite, and schizont gates by dividing the gMFI for each gate by the value measured in the uninfected RBC gate. PfEMP1 expression was assessed using a rabbit polyclonal antibody raised against the N-terminal region (DBLαCIDR didomain) of the predominant PfEMP1 variant expressed in the culture (ITvar70, also known as AFBR6)[52,53]. The staining with PfEMP1 antibody (purified total IgG at 10 µg/ml for 30 mins, followed by APC-conjugated goat anti-rabbit Alexa 647 (Invitrogen, A21244, 2 µg/ml)) was compared to rabbit IgG control antibody (10 µg/ml, total IgG from a non-immunized rabbit, followed by secondary antibody as above). Normalized PfEMP1 gMFI was calculated by subtracting gMFI for rabbit IgG control antibody staining from that of the PfEMP1 antibody staining. Relative PfEMP1 expression for all samples is expressed as a percentage of the average HbAA schizont PfeMP1 expression per experiment, which typically contained three HbAA samples.

**Statistical analysis.** Most data are treated as non-parametrically distributed and presented with medians and interquartile ranges, with the exception of Fig. 5c, where means and standard deviations are shown. Statistical significance was assessed by either two-tailed Mann–Whitney (non-paired data) or two-tailed Wilcoxon signed-rank tests (paired data). Multiple comparisons between stages of RBC infection by *P. falciparum* were analyzed by ANOVA. All calculations were implemented in Prism version 5.04 (GraphPad Software). All p-values quoted for Spearman's rank correlation are two-tailed.

**Reporting summary.** Further information on research design is available in the Nature Research Reporting Summary linked to this article.

## Data availability

Source data can be accessed at https://osf.io/fv6h5/?view_only=65091838d5bb4229b509f 704a1e0c391 (https://doi.org/10.17605/OSF.IO/FV6H5). The mass spectrometry proteomics data have been deposited to the ProteomeXchange Consortium via the PRIDE[54] partner repository with the dataset identifier PXD024022. The corresponding author will reply to inquiries about unprocessed raw data upon reasonable request. Source data are provided with this paper.

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

## Acknowledgements

We are grateful for the assistance provided by both the Microscopy and Histology Core Facility, and the Iain Fraser Cytometry Centre, at the University of Aberdeen. We thank Ann Wheeler and Matt Pearson from Edinburgh Super-Resolution Imaging Consortium for technical support with 3D SIM microscopy. We also thank Janet A. Willment and Bernard Kerscher, supervised by G.D.B., for providing the Fc fusion proteins, Jeanette A. Wagener, supervised by Neil A.R.G. Gow, for providing high purity chitin, Jan Westland for obtaining blood samples and Paul Crocker for useful discussions. Principal funding for this project was provided by Wellcome Trust grant 094847 (R.N.B., L.P.E., M.A.V.). In addition, support was provided by Biotechnology and Biological Sciences Research Council grants BBF0083091 (A.D. and S.M.H.) and BBK0161641 (A.D. and S.M.H.), Wellcome Trust grant 082098 (A.D.), Wellcome Trust grants 97377, 102705 (G.D.B.), and funding for the MRC Centre for Medical Mycology at the University of Aberdeen MR/N006364/1 (G.D.B.).

## Author contributions

H.C. carried out experiments, analyzed data, and wrote the paper. S.H., A.M., B.P., J.S., H.W., M.A.F., E.B., S.L., G.K., B.M., M.-L.W., A. Davie, D.T., M.M., L.H., C.L., W.P. carried out experiments. J.B. supervised by D.C.R., and B.R. obtained blood samples. A.A. obtained and analyzed glycomic and proteomic data, supervised by S.M.H. and A. Dell. L.E., G.D.B., and H.M.W. helped supervise the project. J.A.R. carried out experiments, analyzed data, and wrote the paper. R.N.B. and M.A.V. conceived and supervised the project, and wrote the paper.

## Competing interests

The authors declare the following competing interests: the University of Aberdeen has a patent covering diagnostic and therapeutic applications arising from the work described in this paper (WO/2019/086513). The Wellcome Trust, H.C., R.N.B. and M.A.V. share an interest in this patent. There are no other competing interests.
