## [Peer Review File · Nature Communications]

Reviewers' Comments:

Reviewer #1:

Remarks to the Author:

The presented manuscript describes a novel mechanism by which sickle cell erythrocytes and *P. falciparum* infected HbAS erythrocytes are cleared from circulation. Data presented suggest a common mechanism in that these red blood cells are phagocytosed by macrophages in a process that involves recognition of N-linked high mannose glycans on the surface of red blood cells by the macrophage mannose receptor. The data are interesting and potentially important, and the link between sickle cell disease and protection from malaria by HbAS is exciting. There are, however, a number of issues that require the authors' attention

Points:

1. How does a glycomic analysis of N-glycans from membrane ghosts of HbAA and HbAS erythrocytes look like in comparison to HbSS RBC?
2. The conclusion that a cytoplasmic protein, in this case a component of the membrane skeleton, is N-glycosylated contradicts current views of N-glycosylation and would require further experimental support to be accepted. In particular, one would need to reveal the identity of the modified protein(s) and the exact site in the protein where N-glycosylation occurs.
3. Can the authors exclude the possibility that the GNA signals seen in wild type HbAA red blood cells are associated with remand export vesicles? I find it rather strange that the signal is only seen in a few confined places.
4. In essence, the authors hypothesize that an N-glycosylated membrane skeletal protein changes its location from cytoplasmic to surface exposed. What type of mechanism do the authors envision? The analogy with phosphatidyl-serine is not convincing.
5. How do the authors define a biological replicate? Do they mean an experiment repeated using blood from different donors?
6. A major concern is that the conclusions drawn are mainly based on in vitro experiments and that the link to the in vivo situation is only indirectly established via correlations. It is therefore recommended to validate the key findings in an appropriate animal model system.

Reviewer #2:

Remarks to the Author:

Nature Communications Cao et al.

The concept behind the manuscript is certainly of interest - i.e., that lectins may play a role in removing 'defective' erythrocytes from the circulation. However, the authors are challenging basic concepts about the topology of N-glycosylation and so must be especially careful in the design and interpretation of their experiments.

I assume that the manuscript was originally submitted to Nature, due to the format, also with references in the 'abstract' (also with relatively few citations in total). The last line of the abstract (line 44) is perhaps not strictly correct as 'resistance to malaria infection' is probably rather 'reduced parasitaemia' (as the infection occurs and the infected RBCs are cleared). What we have are correlations, but the 'cell surface' mannose is only shown by the lectin binding (which can anyway be misleading).

The lectin binding is claimed to be specific (on the basis of mannan inhibition - line 64) - what makes it specific should be stated, but of course mannan consists of large glycans, far larger than what is found on RBCs. Although oligomannosidic glycans are found in the 'SCD RBCs', it is a bit misleading not to state also in Line 67 that there is really no difference to the normal RBCs. The use of the word 'mannose' or 'mannose display' is somewhat simplistic - rather oligomannosidic glycans (lines 73, 77, 87). I assume that one usual reason for seeing oligomannosidic glycans (the presence of endoplasmic

reticulum or other membrane-bound organelles) does not apply to mature RBCs.

On line 95 it is then stated that the normal RBCs had not 'exposed' high mannose glycans, which is just seen by lectins. Actually, to give the data a firm basis, a 'cell surface' glycome (e.g., shave intact RBCs with trypsin/PNGase) rather than a 'ghost' glycome should be analysed. It also seems as though only one SCD and one normal ghost glycome was analysed, possibly from pooled material, rather than from different individuals.

A major problem is with the definition of spectrin as being 'decorated with high mannose' (line 107). A band on a blot of a GNA-pull-down recognised by anti-spectrin is not really a proof. If the spectrin molecular weight would be decreased by PNGase F and mass spectrometric analysis would rigorously prove occupation of N-X-S/T sites by N-glycans (e.g., observation of Asn-GlcNAc after EndoH treatment), then the authors would have evidence to overturn the 'prevailing' view about N-glycan topology. Also, if I understand correctly, the authors advocate that the 'mannose' gets translocated to the extracellular side of the membrane - considering that spectrin is a large protein of over 2000 residues, the authors also have to have a mechanism for this - or is some other protein with proven N-glycosylation being translocated instead? Note that retrotranslocation of (especially misfolded) N-glycoproteins into the cytosol can occur as part of degradation, but the RBCs should no longer be synthesising proteins.

The relevance of the chitin inhibition in the phagocytosis assay is not clear (line 130). A further proof of binding of lectins to specific types of RBC glycans would be of a natural/fractionated RBC glycan array - but of course this is not a routine piece of work.

Figure 1 - the glycomes of the normal and SCD RBCs should be compared in the same figure (currently there is not a cross-reference to the Extended Data in the legend).

Figure 3: more controls (PNGase F/EndoH) could be done in order to prove N-glycosylation of spectrin.

Extended Table 2: A listing primarily of proteins with 1 or peptide hits with less than 5% coverage (also of proteins far less than 260 kDa) does not seem to be adequate for inclusion in the table. There should be multiple peptides and a decent coverage (backed up by MS/MS) for inclusion of a certain protein in the list. The significance of the bold annotation is not explained.

Extended Data 2: It seems that 'n' values are missing. Does 'for two HbSS donor RBCs' mean 'for RBCs from two HbSS donors'?

Extended Data 4: the lectin plot in panel b is very faint and 'blot' would be better as a description as not just GNA is being used. Also, the 'window' of the shown blot is very restrictive (as compared to Figure 3) - the anti-spectrin signal also increases upon PNGase/Endo treatment - it should remain the same (any planned exaggeration should be stated). The difference in magnification of the spectra should be stated; also if there are differences in annotation (e.g., the structures at 3025 or 3473 are not annotated consistently and 3473 is also the same composition as 3474 in Figure 1), are these based on different MS/MS or is just a random drawing of what fits the mass? I note there is no glycomic analysis of malaria-infected RBCs or of 'oxidatively-stressed' RBCs. Longer LacNAc-based chains are, of course, well known for RBCs (e.g., defective in HEMPAS); if there is information regarding the ABO blood group of the donors, can this be seen by the MS/MS data or are the potential antennal fucose residues Lewis-type?

Note that my comments are primarily on the sugar aspects, as requested by the journal.

Reviewer #3:
Remarks to the Author:

The main claims of the manuscript are

- 1) Red blood cells (RBC) from sickle cell disease (SCD) patients display mannose-rich structures at the cell surface. These structures also occur after oxidative stress and malaria infection. Importantly SCD RBCs display more mannose-rich structures after malaria infection than healthy RBC.
- 2) Data are consistent with the RBC protein spectrin displaying N-linked high mannose structures in untreated and oxidized RBC with mannosylation slightly increased after oxidation. These results are in agreement with the fact that intracellular mannose structures are already present in healthy RBC and that RBC stress, as seen in SCD, oxidation and malaria infection, promotes extracellular exposure.
- 3) Mannosylated structures are selectively recognised by the MR-CTLD region
- 4) Human macrophages preferentially phagocytose oxidised RBC and this uptake is reduced by the presence of Mannan, chitin (MR ligands) and anti-MR antibodies, and inhibition of MR expression using siRNA.
- 5) Finally, the authors provide a vast range of clinical data that support the clinical relevance of their findings.

The results are novel and of general interest. They improve understanding of the pathogenesis of the sickle cell disease and the protective phenotype of HbS heterozygosity in malaria. My comments mostly relate to data interpretation.

A) Authors state in their manuscript that splenic macrophages are involved in the phagocytosis of SCD and malaria-infected RBCs. I would invite the authors to consider the important differences in splenic architecture between mouse and human spleen [Science Immunology 01 Mar 2019: Vol. 4, Issue 33, eaau6085 DOI: 10.1126/sciimmunol.aau6085] and the fact in human spleen that mannose receptor is not expressed by macrophages but by cells lining the venous sinuses [Laboratory Investigation volume 85, pages 1238–1249 (2005)].

B) Based on later results (Extended data Fig 4), glycomic analysis of SCD and Healthy controls RBCs both show presence of mannosylated glycans. Hence I do not think it is appropriate to highlight the presence of mannosylated ligands in SCD RBC in the main text. Also I am not quite sure why authors state that N-linked high mannose glycans are “abundant”. Their abundance does not seem that impressive which would be in agreement with their restricted location.

C) In figure 2f it is not clear what each data point refers to. Further explanation is required.

D) To better illustrate exposure of mannosylated ligands only in stressed cells, the authors should show side by side analysis of permeabilised and non permeabilised healthy and SCD RBC including actin labelling: Healthy RBC should bind mannose-specific lectins only when permeabilised (actin+), while SCD RBC should bind both when permeabilised (actin+) and non permeabilised (actin -). The authors should provide a video showing 3D images of the spectrin co-localisation with mannosylated structures to better validate the characterisation of mannosylated spectrin as the culprit of the binding of mannose-specific lectins. In addition, spectrin and a control RBC protein should be immunoprecipitated from RBC lysates and tested for GNA binding.

E) The MR labelling shown in Fig 3i is unconvincing (it looks cytosolic rather than endosomal in some cells) and it is unclear if MR co-localises with RBC-containing phagosomes. It would be of interest to know if MR is recruited to the phagocytic cup.

F) In Fig 3 k-n the authors should explain what each data point refers to. Figure legend mentions “derived from 3 experiments” but further explanation is required.

G) Data presentation in Figure 4A is confusing.

H) In methodology it is unclear why in Donors section samples were collected in EDTA (EDTA what?) and in RBC isolation in ACD solution.

I) Validation of siRNA should include a test to determine that other receptors are not affected.

Referee 1 describes the work as “interesting and potentially important, and the link between sickle cell disease and protection from malaria by HbAS is exciting”.

1. How does a glycomic analysis of N-glycans from membrane ghosts of HbAA and HbAS erythrocytes look like in comparison to HbSS RBC?

The glycomic analyses from HbSS (top) and HbAA (bottom) ghosts are very similar. All five members of the high mannose family and complex N-glycans, most of which are sialylated, are seen in both. Perhaps there are more high mannoses in the SS ghosts, but these analyses are only semi-quantitative and we do not place a great deal of reliance on this conclusion. As there are no systematic differences between HbAA and HbSS ghosts, we did not extend the study to HbAS erythrocytes.

Sickle cell ghost N-glycome: High mannose 48.3%, Complex N-glycan 51.7%

Healthy cell ghost N-glycome: High mannose 35.9%, Complex N-glycan 64.1%

2. *The conclusion that a cytoplasmic protein, in this case a component of the membrane skeleton, is N-glycosylated contradicts current views of N-glycosylation and would require further experimental support to be accepted. In particular, one would need to reveal the identity of the modified protein(s) and the exact site in the protein where N-glycosylation occurs.*

In the manuscript submitted last year, we did not claim that spectrin was directly N-glycosylated. We were careful not to mention spectrin in the title or abstract and the only phrase we used in the main text was ‘spectrin-containing complex is decorated with high mannose’. We did, however, present several pieces of evidence, from both blots and microscopy, showing associations between spectrin and high mannose N-glycans.

Prompted by the referees’ comments, in the last year we have obtained more extensive data supporting the associations between spectrin and high mannoses. This association appears to be covalent, as they are inseparable by any detergent or chaotropic agent that we used.

We started by repeating the original straightforward approach of cutting out the main GNA lectin binding bands, digesting with trypsin and analysing using conventional mass spectrometric glycoproteomics. This gave similar results to our first attempt: identification of numerous tryptic peptides from spectrin as well as an admixture of proteins with lower molecular weights, but no glycopeptides. As will become clear, we believe the lack of identifiable individual glycopeptides has a significance we did not originally fully appreciate.

As a new strategy, we performed partial trypsin digestions of healthy (HbAA) red blood cell ghosts and found matching patterns of α -spectrin and GNA lectin staining among the fragments. Sickle cells, however, naturally harboured lower molecular weight GNA lectin binding bands without trypsin treatment. These were consistently endoglycosidase-H (EndoH) sensitive, indicating high mannose glycosylation, and the intensity of one such band correlated with cell surface GNA lectin binding. Similar bands could also be induced by aging of HbAA cells, indicating that they could be produced by proteolysis. The sickle GNA lectin binding fragments exhibited resistance to digestion by several proteases and, paradoxically, increased in intensity with longer digestions, indicating generation from a higher molecular weight source that could not be seen on blots. In the manuscript and below, we present evidence that this source comprises high molecular weight complexes of damaged proteins, which includes spectrin. In brief, the lines of evidence indicating an atypical structure are: protease resistance, partial resistance to glycosidases, presence of cryptic epitopes, presence of glycated peptides and difficulties in identification of peptides by conventional mass spectrometric approaches.

3. *Can the authors exclude the possibility that the GNA signals seen in wild type HbAA red blood cells are associated with remand export vesicles? I find it rather strange that the signal is only seen in a few confined places.*

This turned out to be a perceptive observation/question. In the new manuscript, we now include additional super-resolution microscopy images. At high magnification using 3D SIM (Fig. 4f), isolating a single sheet of the red cell membrane, clusters of GNA lectin binding can be seen on the HbAA spectrin network. 3D SIM (Fig. 4j) shows that HbSS RBC surface GNA lectin binding corresponds to underlying large aggregates of spectrin, whose significance we did not initially appreciate.

Autophagic export vesicles are described in sickle cell disease (Mankelov et al. Blood. 2015;126(15):1831-1834. doi:10.1182/blood-2015-04-637702). However, these are phosphatidylserine positive and have a different morphology to the structures we describe. We also investigated whether our new structures corresponded to ‘pits’ that result from another export process and which are known to be increased in number in sickle cell disease, but they are also different structures. We have not included these details in the resubmitted manuscript.

4. *In essence, the authors hypothesize that an N-glycosylated membrane skeletal protein changes its location from cytoplasmic to surface exposed. What type of mechanism do the authors envision? The analogy with phosphatidyl-serine is not convincing.*

We did not mean to imply that the mechanism for the externalisation was similar to that described for phosphatidylserine flipping, merely highlighted the topological analogy, which we have now removed. The mechanism whereby high mannoses become available for binding by extracellular ligands is not directly addressed in our paper and we thus have not emphasized the issue in the manuscript. We did, however, notice one of the lower molecular weight fragments (100kDa) shows a good correlation with surface GNA lectin binding. We speculate that there may be a specific proteolytic cleavage of spectrin involved in a translocation across the plasma membrane and this possibility is alluded to in the manuscript. Furthermore, spectrin has numerous phosphatidylserine binding domains and there is a close relationship between the protein and the lipid bilayer. The plasma membrane exists in a fluid phase and, if anchored on either side, it is easy to envisage a loop could be inserted through the membrane. As we have no direct evidence for this possibility, we do not mention it in the paper. It is also possible that the glycosylated residues derive from other degraded membrane proteins. In this case, there is no need to provide a mechanism for spectrin itself to cross the plasma membrane.

5. *How do the authors define a biological replicate? Do they mean an experiment repeated using blood from different donors?*

Yes, we mean blood from different donors.

6. *A major concern is that the conclusions drawn are mainly based on in vitro experiments and that the link to the in vivo situation is only indirectly established via correlations. It is therefore recommended to validate the key findings in an appropriate animal model system.*

We agree that it would have been better if we had replicated some these findings *in vivo*. To this end, we initiated a study in mice, but found there were several differences between mice and men that would mean replication would have only limited validity. In retrospect, perhaps this should not have been surprising. Most importantly, there is a fundamental difference between mice and humans in that in humans the main organ for red cell removal is the spleen, whereas it is the liver in mice. Murine red blood cells have a lifespan of 40 days versus 120 days in humans. Furthermore, as highlighted by referee 3, the cellular distribution of the mannose receptor within the spleen is very different between the two species. Finally, the structural features of N-glycans show several differences between the two species.

Indeed, when we started to study murine red cells, we found that some terminal mannoses were expressed on their red cell surface constitutively. Also, lectin blots from murine ghosts showed several bands rather than just spectrin. While it would be very interesting to elucidate the structural and physiological bases for these differences, we felt the reason we were looking was to replicate the situation in humans and this was clearly not going to be the case. We therefore terminated these studies and decided to concentrate our efforts on expanding the range of human sickle cell disease studied, which provided great insights in an *in vivo* setting.

It is also interesting to speculate on why these profound differences arose, when most physiological systems studied are more similar between the two species. Rapid evolution most commonly happens under selective pressure from infections. *Plasmodium* species infect most primates and most of human evolution occurred in Africa, where malaria is endemic. *P. falciparum* has had a profound impact on the human genome, with at least one third of humans worldwide carrying mutations known to protect against severe malaria. The sickle mutation itself has been modelled to provide a 10% increase in survival rate per generation since arising about 6000 years ago. We therefore speculate that primates developed these differences in order to enhance their immune responses against infections with plasmodia by switching their red cell disposal mechanisms to an organ with a better system of ‘filtration’. However, we feel these thoughts are too speculative to include in the Discussion of the paper itself.

Referee 2 states the concept is “certainly of interest” and acknowledges we are ‘challenging basic concepts about the topology of N-glycosylation’ but cautions we must therefore be especially careful in the design and interpretation of experiments.

The general issues concerning the nature of N-glycosylation are discussed above. In neither the original submission nor the resubmission do we claim that spectrin itself is directly N-glycosylated.

The last line of the abstract (line 44) is perhaps not strictly correct as ‘resistance to malaria infection’ is probably rather ‘reduced parasitaemia’ (as the infection occurs and the infected RBCs are cleared). What we have are correlations, but the ‘cell surface’ mannose is only shown by the lectin binding (which can anyway be misleading).

We now use the wording ‘resistance to severe malaria’ in the abstract.

In the last year we have obtained extensive new data from sickle red cells, all supporting the conclusion they are characterised by high mannoses available for binding on their surfaces. Extended Data Table 1 lists possible artefacts that might explain the high levels of mannose specific binding observed and their exclusion. We now present the existence of several bands binding GNA lectin of lower molecular weight than full length spectrin in ghosts from sickle cell ghosts (Fig. 4a). Numerous blots show that these bands, both naturally occurring and produced by proteases, co-stain with anti-spectrin antibodies (Extended Data Figure 6a, b, Extended Data Fig. 7d, e). The intensities of one of these bands correlates with surface GNA lectin binding (Fig. 4b). Binding of GNA lectin to all these bands is PNGase/EndoH sensitive (Figure 4c, 4d, Extended Data Figs. 6d, 7c). EndoH is also able to decrease the signal on whole sickle cells (see below). We also present data showing that these bands can be

generated by proteolysis (Fig. 4g, Extended Data Fig. 6e, f, g), implicating high molecular weight complexes as sources of these bands. Finally, we characterise the most protease resistant band in detail and show it contains both high mannoses and a portion of α -spectrin. It also exhibits several properties (poor representation of peptides in proteomic analysis, protease resistance, requirement of denaturing conditions for EndoH digestion, possession of cryptic epitopes and glycated amino acids) indicating an unusual peptide structure, probably arising from damage to amino acids.

The lectin binding is claimed to be specific (on the basis of mannan inhibition - line 64) - what makes it specific should be stated, but of course mannan consists of large glycans, far larger than what is found on RBCs. Although oligomannosidic glycans are found in the 'SCD RBCs', it is a bit misleading not to state also in Line 67 that there is really no difference to the normal RBCs. The use of the word 'mannose' or 'mannose display' is somewhat simplistic - rather oligomannosidic glycans (lines 73, 77, 87). I assume that one usual reason for seeing oligomannosidic glycans (the presence of endoplasmic reticulum or other membrane-bound organelles) does not apply to mature RBCs.

We admit that mannan, a linear mannose polymer, is not as good a competitive inhibitor as would high mannoses themselves be, but it does have the advantage of being available for purchase and is widely used in this role. But this is not the only reason for our claims of specificity. Supportive evidence is also provided by the competitive blockers targeted at the mannose receptor (chitin and a blocking antibody). Further specificity is provided by our demonstration that GNA lectin binding is inhibited on numerous occasions by incubation with PNGase and Endo-H, the glycomic analyses we present, as well as reasons listed in Extended Data Table 1.

The differences between high mannoses seen in glycomic analyses of HbAA versus HbAS cells are discussed above.

We concede the use of the terms 'mannose' and 'mannose display' was a rather loose, although does have the advantage of brevity. However, we think the term 'oligomannosidic glycans' is too general. High mannoses were demonstrated in the whole ghost glycomics and the glycomic analysis from the protease resistant fragment F40 (see Figure 4h and Ext. Data Fig 7b). The tri-mannose structure cannot be the source of the GNA lectin binding as the binding is Endo-H sensitive and the enzyme reacts with Man5(+fuc) and Man5-Man9 structures but not Man3(+Fuc). 'High mannose' is therefore a more accurate term rather than 'oligomannosidic glycans'. We have eliminated use of 'mannose' or 'mannose display' and standardised on 'high mannose N-glycans'.

The reviewer is correct to state that mature RBCs do not contain ER/Golgi. Indeed, this new mechanism would have been difficult to detect in a nucleated cell where any high mannoses detected would have been ascribed to the presence of ER/Golgi.

On line 95 it is then stated that the normal RBCs had not 'exposed' high mannose glycans, which is just seen by lectins. Actually, to give the data a firm basis, a 'cell surface' glycome (e.g., shave intact RBCs with trypsin/PNGase) rather than a 'ghost' glycome should be analysed. It also seems as though only one SCD and one normal ghost glycome was analysed, possibly from pooled material, rather than from different individuals.

We did not pool material from multiple individuals but analysed several healthy, and a sickle donor, glycomes individually. We also performed surface trypsin and Endo-H digestions with intact RBCs and demonstrated reductions in GNA lectin binding:

Endo-H cell surface treatment causes a loss of approximately one quarter of the differential GNA staining between HbSS and HbAA red cells, supporting the presence of high mannoses on the surface of sickle cells. In the new manuscript (Fig. 4h), we show that for some structures, denaturing conditions are required for full Endo-H sensitivity. Since this Endo-H treatment is performed in a cell friendly, non-denaturing environment, full loss of signal is not surprising.

Conversely, incubation of intact cells with trypsin, caused an increase in GNA binding. This would be consistent with an unmasking of high mannose signals through proteolytic cleavage in a similar way that proteolytic cleavage allowed unmasking of a cryptic epitope in Extended Data Fig. 7d+e. In the new manuscript, we show that proteins carrying high mannose glycans are frequently resistant to proteolysis by trypsin. Due to limitations of space, we have not included these data in the new manuscript, but could do so if requested.

A major problem is with the definition of spectrin as being ‘decorated with high mannose’ (line 107). A band on a blot of a GNA-pull-down recognised by anti-spectrin is not really a proof. If the spectrin molecular weight would be decreased by PNGase F and mass spectrometric analysis would rigorously prove occupation of N-X-S/T sites by N-glycans (e.g., observation of Asn-GlcNAc after EndoH treatment), then the authors would have evidence to overturn the ‘prevailing’ view about N-glycan topology. Also, if I understand correctly, the authors advocate that the ‘mannose’ gets translocated to the extracellular side of the membrane - considering that spectrin is a large protein of over 2000 residues, the authors also have to have a mechanism for this - or is some other protein with proven N-glycosylation being translocated instead? Note that retrotranslocation of (especially misfolded) N-glycoproteins into the cytosol can occur as part of degradation, but the RBCs should no longer be synthesising proteins.

As discussed above, we are not claiming that spectrin is directly N-glycosylated. We now provide multiple lines of evidence that spectrin is associated with an unusual glycosylated structure: evidence of an appropriate shift in molecular weight with glycosidase treatment (Fig. 4d), numerous examples of coincidence of bands using GNA lectin and anti-spectrin antibodies (Extended Data Fig. 6a, b, Extended Data Fig. 7d, e), further GNA pull-downs (Extended Data Fig. 6d), as well as glycoproteomic analysis of a protease resistant band (F40: Figure 4g-I, Extended Data Fig. 7b, c).

Mechanisms of how high mannoses might become available for recognition are discussed in the reply to referee 1 above.

The relevance of the chitin inhibition in the phagocytosis assay is not clear (line 130). A further proof of binding of lectins to specific types of RBC glycans would be of a natural/fractionated RBC glycan array - but of course this is not a routine piece of work. Chitin is known to block the mannose receptor and the following phrase (Line 120-121) clarifies this: ‘.....phagocytosis was also inhibited by the competing glycans mannan or chitin, each known to block MR-CRD.....’

Figure 1 - the glycomes of the normal and SCD RBCs should be compared in the same figure (currently there is not a cross-reference to the Extended Data in the legend).

We present the two glycomes together in the answer to Reviewer 1 (Q1). However, we present the two glycomes separately in the new manuscript, as we feel this fits the presented narrative better. We now include a cross-reference in the legend to Fig. 1 ‘(A similar glycome from and HbAA donor is shown in Extended Data Fig. 4.)’.

Figure 3: more controls (PNGase F/EndoH) could be done in order to prove N-glycosylation of spectrin.

More relevant data are now provided in Fig. 4 and Extended Data Figs. 6 and 7.

Extended Table 2: A listing primarily of proteins with 1 or peptide hits with less than 5% coverage (also of proteins far less than 260 kDa) does not seem to be adequate for inclusion in the table. There should be multiple peptides and a decent coverage (backed up by MS/MS) for inclusion of a certain protein in the list. The significance of the bold annotation is not explained.

We have now omitted all entries with only a single peptide. In light of the unusual biochemical properties of the high mannose decorated protein aggregates, this list indicates putative cross-linked peptides, although we have not been explicit about this possibility in the manuscript.

Extended Data 2: It seems that 'n' values are missing. Does 'for two HbSS donor RBCs' mean 'for RBCs from two HbSS donors'?

The samples were from different individuals. At no point in this project was any sample pooled with another.

Extended Data 4: the lectin plot in panel b is very faint and 'blot' would be better as a description as not just GNA is being used. Also, the 'window' of the shown blot is very restrictive (as compared to Figure 3) - the anti-spectrin signal also increases upon PNGase/Endo treatment - it should remain the same (any planned exaggeration should be stated).

We now present higher quality lectin blots showing PNGase/EndoH digests concentrating on the lower molecular weight bands (Figure 4, Extended Data Figs. 6, 7). In fact, we have found glycosidase sensitivity at 260kDa, although often demonstrable, is inconsistent, and so have omitted these data in the new manuscript.

The difference in magnification of the spectra should be stated; also if there are differences in annotation (e.g., the structures at 3025 or 3473 are not annotated consistently and 3473 is also the same composition as 3474 in Figure 1), are these based on different MS/MS or is just a random drawing of what fits the mass? I note there is no glycomic analysis of malaria-infected RBCs or of 'oxidatively-stressed' RBCs. Longer LacNAc-based chains are, of course, well known for RBCs (e.g., defective in HEMPAS); if there is information regarding the ABO blood group of the donors, can this be seen by the MS/MS data or are the potential antennal fucose residues Lewis-type?

We are unclear what is meant by 'magnification of spectra'. The annotations have now been made consistent. We have analysed the glycomes of RBCs that have been aged for 60 days, calcium ionophore treated (eryptosis) and oxidation (copper sulphate/ascorbic acid). As seen in the below spectra, the high mannoses are present in all five forms (Man₅₋₉GlcNAc₂), with the oxidation inducing a slight skew of the overall N-glycans towards high mannoses. However, upon multiple repeats, we do not think this skewing is significantly different. These analyses are only semi-quantitative nature and we have not included them in the accompanying manuscript. In view of the lack of consistent differences, we decided not to analyse the glycome of malaria infected RBCs.

Blood groups A and B can be identified from MS/MS analyses, because the structures carrying these epitopes exhibit characteristic fragment ions (m/z 905 and 864, for blood groups A and B respectively). The glycomic analysis from the healthy sample was found to contain blood group A. Conversely, blood group O, where the fucose residue is attached to the terminal galactose residue, does not exhibit a characteristic fragment that allows its discrimination from other epitopes such as Lewis-X/A (all common m/z 660). Our glycomic analyses derive from different donors. For this reason, the antenna fucose residues are outside the bracket. However, MALDI-TOF/TOF MS/MS analyses indicated that these epitopes were not Lewis-X. The purpose of the glycomic analysis was the characterization of the high mannose N-glycans since the complex N-glycans do not seem to participate in the mechanism of interest.

Structural analysis of complex N-glycans derived from the SCD patient. MALDI-ToF/ToF MS/MS spectra (m/z versus relative intensity) of the molecular ions at (a) m/z 3286, (b) m/z 3561, (c) m/z 4185 and (d) m/z 4808 selected from membrane ghosts of HbSS. All molecular ions are [M+Na]⁺. Horizontal blue dashed lines with arrowheads indicate

losses of the corresponding structures from the molecular ions. Structures outside a bracket have not had their location unequivocally defined. Putative structures are based on composition, tandem MS and knowledge of biosynthetic pathways. Major structures are shown. Annotation uses conventional symbols for carbohydrates in accordance with <http://www.functionalglycomics.org> guidelines: purple diamond, sialic acids; yellow circle, galactose; blue square, *N*-acetyl glucosamine; green circle, mannose; red triangle, fucose. Note the following: (i) the fragment ions at *m/z* 3080, 3355, 3979 and 4602 corresponding to the elimination of fucose residue are not found on **a**, **b**, **c** and **d** respectively. These fragment ions, when present, are indicative of fucose residues being in α 1,3 linkage. Therefore, data indicate that the terminal epitopes in HbSS are not of Lewis-X epitopes. (ii) In high-mass N-glycans (ex. molecular ions at *m/z* 4185 and 4808), the antennas are extended as mixtures of linear and I-branched LacNAcs.

Referee 3 states the results are novel and of general interest. They improve understanding of the pathogenesis of the sickle cell disease and the protective phenotype of HbS heterozygosity in malaria.

A) Authors state in their manuscript that splenic macrophages are involved in the phagocytosis of SCD and malaria-infected RBCs. I would invite the authors to consider the important differences in splenic architecture between mouse and human spleen [Science Immunology 01 Mar 2019: Vol. 4, Issue 33, eaau6085 DOI: 10.1126/sciimmunol.aau6085] and the fact in human spleen that mannose receptor is not expressed by macrophages but by cells lining the venous sinuses [Laboratory Investigation volume 85, pages 1238–1249 (2005)].

We thank the referee for these pointers. We were particularly interested in the latter study showing that the mannose receptor is expressed by specialised ‘Lyve-1+ cells lining venous sinuses, where they form a physical barrier for blood cells to exit the red pulp and so are ideally located to perform a filtering function’.

B) Based on later results (Extended data Fig 4), glycomic analysis of SCD and Healthy controls RBCs both show presence of mannosylated glycans. Hence I do not think it is appropriate to highlight the presence of mannosylated ligands in SCD RBC in the main text. Also I am not quite sure why authors state that N-linked high mannose glycans are “abundant”. Their abundance does not seem that impressive which would be in agreement with their restricted location.

We make it clear in the new manuscript that high mannoses are found in both HbAA and HbAS RBCs, but that they are only available as ligands for extracellular receptors on HbSS cells. We agree that we should not have used the term abundant in the absence of a quantitative measure and have removed the term in this context.

C) In figure 2f it is not clear what each data point refer to. Further explanation is required.

Each data point refers to a different RBC donor, which we have clarified in the new figure legend.

D) To better illustrate exposure of mannosylated ligands only in stressed cells, the authors should show side by side analysis of permeabilised and non permeabilised healthy and SCD

RBC including actin labelling: Healthy RBC should bind mannose-specific lectins only when permeabilised (actin+), while SCD RBC should bind both when permeabilised (actin+) and non permeabilised (actin -). The authors should provide a video showing 3D images of the spectrin co-localisation with mannose-specific lectins to better validate the characterisation of mannose-specific lectins as the culprit of the binding of mannose-specific lectins. In addition, spectrin and a control RBC protein should be immunoprecipitated from RBC lysates and tested for GNA binding.

We observe essentially no binding of GNA lectin to the surfaces of intact healthy cells and punctate staining inside healthy permeabilized cells. In the manuscript, the comparison between surface staining of HbAA and HbSS cells is shown in Fig. 1b, colocalization between GNA lectin and spectrin staining in permeabilized cells in Fig. 4f and Extended Data Fig. 4b, and colocalization between surface GNA lectin staining and spectrin staining in subsequently permeabilized cells in Fig. 4j. In Fig. 4f and Extended Data Fig. 4b, GNA lectin and spectrin staining to permeabilized HbAA and HbSS RBCs again show the punctate nature of the stained structures. At super-resolution, we see the appearance of GNA binding only on a small population of spectrin. In addition, we show further images confirming these findings below. Both oxidized healthy cells and non-oxidized sickle cells show punctate surface staining without permeabilization. Permeabilization allows staining of healthy cells and is verified with actin staining (green).

E) The MR labelling shown in Fig 3i is unconvincing (it looks cytosolic rather than

endosomal in some cells) and it is unclear if MR co-localises with RBC-containing phagosomes. It would be of interest to know if MR is recruited to the phagocytic cup.

We were in the process of growing macrophages to address this point when the lockdown came and we had to discard all our cultures. We hope you understand this was beyond our control.

F) In Fig 3 k-n the authors should explain what each data point refers to. Figure legend mentions “derived from 3 experiments” but further explanation is required.

We have now a general statement defining what each data point refers to.

G) Data presentation in Figure 4A is confusing.

Now made clearer in both y-axis and legend.

H) In methodology is unclear why in Donors section samples were collected in EDTA (EDTA what?) and in RBC isolation in ACD solution.

Clinical samples for haematological analysis are routinely collected into ‘EDTA tubes’ (Becton-Dickinson), or more accurately dipotassium ethylenediaminetetraacetic acid, which chelates calcium to prevent coagulation and preserves cellular morphology, at the cost of cellular viability. In the university, for uptake experiments, gentler calcium chelator acid dextrose dextrose (ACD) solutions were used to preserve monocyte viability. The nature of the tubes has been made more explicit in the manuscript.

I) Validation of siRNA should include a test to determine that other receptors are not affected.

HLA-DR was imaged with the same MR siRNA and scramble siRNA. Staining remains unchanged. We were in the process of extending the range of receptors tested when the lockdown came.

CD206 SiRNA (HLA-DR intracellular staining)

-ve control SiRNA (HLA-DR intracellular staining)

Reviewers' Comments:

Reviewer #1:

Remarks to the Author:

The authors have adequately addressed my comments.

Michael Lanzer

Reviewer #2:

Remarks to the Author:

Nature Communications Cao et al. revised

The authors have, after over a year, resubmitted their manuscript. They have done a significant rewrite and have added more data. Seemingly, I was not the only referee to be concerned/confused about the N-glycan topology issue – but the first version was somewhat unclear. However, in my honest opinion, the authors have not explained their phenomenon – in terms of neither the “exposed” mannose nor the relevant glycoprotein. As stated in my original review, the overall idea about the study is of interest.

Lines 40-53: The introduction is rather short and HbA/HbF are not explained.

Lines 62-63 and 331-337: Species names are written with like *Galanthus nivalis* (genus with capital, species with small first letter).

Lines 68-70: This sentence is a bit misleading as also normal RBCs have lots of high mannose, not just the SCD RBCs.

Line 80: The authors could add – “Despite a similar glycomic profile, RBCs from patients with SCD ...”

Line 83: high levels of “exposed” mannose

Line 118: knockdown of MR “in macrophages”

Line 129 ff: If reducing SDS-PAGE was employed, then it must be a highly unusual complex to survive SDS, unless there is a covalent bond involved (as mentioned in reply). If a short glycopeptide has been oxidatively cross-linked to spectrin, then tryptic peptide mapping will probably not help as potentially the cross-linked peptide is not ‘released’. Other mass spectrometric approaches are still necessary. Oxidation of lysines resulting in inability to cleave with trypsin means that, e.g., GluC could be attempted. Anyway, in the end the glycoprotein/glycopeptide component has not been identified.

Figure 1c: It should be stated that the spectrum is from “one” HbSS donor and, for the sake of the reader, that a ‘normal’ spectrum should be presented alongside it, rather than having the ‘similar’ normal glycome shown in the Extended data. How many normal and ‘abnormal’ glycomes were analysed? If the 48% for the SCD glycome and the 36% for the normal are ‘usual’ percentages (see reply; not mentioned in manuscript), then the higher percentage high mannose in the SCD RBCs becomes significant, but statistics on multiple donors should be included. Also, the question is whether the RBCs were trypsinised before glycan release could be answered in a short statement in the methods; it becomes relevant in the context that tryptic mapping failed to result in identification of the spectrin-associated glycopeptide.

Figure 2a and Extended Data 3: State the n number in the legend.

Figure 5: why not have data for HbSS also in panel a?

Extended Data 4: include the overall spectrum in Figure 1; it should be stated that these are the glycans from one HbAA donor. The zoom factor for the 'high mass' spectrum should be indicated (also for Figure 1c) so that readers can assess also the low abundance glycans. The annotations differ between the SCD and normal glycomes – e.g., are there branched LacNAc or multiantennary glycans? (Or is there ambiguity/mixture?)

Extended Data 7: Some major peaks in panel b are not identified – if impurities, then this should be stated.

The full mass spectra should be uploaded in 'searchable' form (Excel, mzxml or something similar) or larger sized MS data figures included in the Extended Data.

Reviewer #3:

Remarks to the Author:

The authors provide a revised version of the manuscript that includes a substantial amount of new data on the biochemical characterisation of the mannose-containing ligands exposed in HbSS, oxydised, infected and aged erythrocytes. I still think this is a relevant and important manuscript and, in my view, some of the additional data improve the quality of the work. I strongly sympathise with the research team as they worked very hard to address this challenging problem.

First, I wanted to highlight that I found the new proteomics sections and associated figures very difficult to follow. I believe I got a general idea of what has been done and what this means but I am not sure I got all the details right. I would recommend rewriting the text and the inclusion of cartoons describing the experimental process and better annotation of the figures. There are instances of mislabelling. In Figure 4 the legend does not seem to correspond to the data shown; section b does not show "Surface GNA lectin staining by flow cytometry (normalized gMFI)(bottom) and 844 corresponding GNA westerns (top) from HbAS and HbSS ghosts. (Right) Spearman's 845 rank correlations between FACS data and band intensities, both classified ordinally as 846 high, medium or low (n=27 measurements from 22 individuals)."

Other points are:

Based on the new findings, proteolysis of spectrin in response to stress signals generates low molecular weight bands that display the sugars. Based on this I would expect that pull down of GNA-binding material from HbSS ghosts would show low molecular weight bands recognised by the anti-spectrin antibody. This would agree with the western blot data from Extended Figure 6.

Figure 4e includes a pull down of GNA-binding material from healthy erythrocytes and shows a single high molecular weight band recognised by the anti-spectrin antibody which indicates that spectrin might be part of a complex that contains high mannose structures in healthy cells. Based on Figure 4f, these structures are present intracellularly in HbAA red blood cells but are more abundant and closer to the surface in HbSS red blood cells. Based on the images, it seems that the cellular distribution of spectrin differs between HbAA and HbSS red. Is this a possibility? 3D high resolution microscopy results (Figure 4j) are consistent with spectrin being associated with high mannose patches displayed at the cell surface in HbSS erythrocytes.

I was very intrigued by the formation of high molecular complexes that include spectrin. This was discussed in the manuscript but going through the literature I found the manuscripts below that describe cross-linking of spectrin and haemoglobin linked to senescence and oxidative damage and I wonder if the authors, by limiting the analysis to material that enters a gel, might be missing some important players and if they could detect some haemoglobin in their preparations.

Irreversible spectrin-haemoglobin crosslinking in vivo: a marker for red cell senescence (1983)

L. M. Snyder L. Leb J. Piotrowski N. Sauberman S. C. Liu N. L. Fortier

<https://doi.org/10.1111/j.1365-2141.1983.tb02038.x>

Effect of hydrogen peroxide exposure on normal human erythrocyte deformability, morphology, surface characteristics, and spectrin-hemoglobin cross-linking.

L M Snyder, ... , S Shoet, N Mohandas

J Clin Invest. 1985;76(5):1971-1977. <https://doi.org/10.1172/JCI112196>.

Thank you again for submitting your manuscript "High mannose N-glycans on red blood cells as phagocytic ligands, mediating both sickle cell anaemia and resistance to malaria" to Nature Communications. We have now received reports from 3 reviewers and, on the basis of their comments, we have decided to invite a revision of your work for further consideration in our journal. Your revision should address all the points raised by our reviewers (see their reports below). In particular, a revised manuscript will need to provide statistical analysis of glycomes of multiple donors to validate results along the lines suggested by reviewer #2, describe the proteomics in more detail, and appropriately address all other concerns from our reviewers.

We thank the editor and referees for their further comments. An important point highlighted by the editor was a statistical comparison between glycomes from donors with sickle cell disease versus healthy controls. We have performed further glycomic analyses in order to provide data from additional donors to allow a statistical comparison (below). The second are highlighted was the proteomic description, which has been addressed with an extensive rewrite. Individual comments are addressed below. Changes in the revised manuscript are highlighted in green.

REVIEWER COMMENTS

*Reviewer #1 (Remarks to the Author):
The authors have adequately addressed my comments.
Michael Lanzer*

We thank the referee for his acknowledgment of our changes.

*Reviewer #2 (Remarks to the Author):
Nature Communications Cao et al. revised
The authors have, after over a year, resubmitted their manuscript. They have done a significant rewrite and have added more data. Seemingly, I was not the only referee to be concerned/confused about the N-glycan topology issue – but the first version was somewhat unclear. However, in my honest opinion, the authors have not explained their phenomenon – in terms of neither the “exposed” mannose nor the relevant glycoprotein. As stated in my original review, the overall idea about the study is of interest.*

We are pleased that the referee agrees that the study is of interest, and believe that we have addressed the issue of the precise associations of the high mannose glycans as fully as is currently possible in the revised paper, which focuses on the novelty of their exposure and biological effects.

Lines 40-53: The introduction is rather short and HbA/HbF are not explained.

The physiology and pathology of haemoglobin molecules and their mutations are complex subjects. We originally presented a simple approach by using only the terms HbA, HbS and HbF, on the basis that a fuller understanding doesn't necessarily add to the main messages of the paper. We now include a more detailed explanation of haemoglobin chains and variants to help with readers' understanding of the patient derived data.

Lines 62-63 and 331-337: Species names are written with like Galanthus nivalis (genus with capital, species with small first letter).

All instances changed.

Lines 68-70: This sentence is a bit misleading as also normal RBCs have lots of high mannose, not just the SCD RBCs.

We now present an HbAA glycomic profile alongside that of HbSS, have added a sentence about healthy RBC having also high mannose glycans and include a statistical comparison between the two.

Line 80: The authors could add – “Despite a similar glycomic profile, RBCs from patients with SCD ...”

We have incorporated this suggestion.

Line 83: high levels of “exposed” mannose

We have incorporated this suggestion.

Line 118: knockdown of MR “in macrophages”

We have incorporated this suggestion.

Line 129 ff: If reducing SDS-PAGE was employed, then it must be a highly unusual complex to survive SDS, unless there is a covalent bond involved (as mentioned in reply). If a short glycopeptide has been oxidatively cross-linked to spectrin, then tryptic peptide mapping will probably not help as potentially the cross-linked peptide is not 'released'. Other mass spectrometric approaches are still necessary. Oxidation of lysines resulting in inability to

cleave with trypsin means that, e.g., GluC could be attempted. Anyway, in the end the glycoprotein/glycopeptide component has not been identified.

We agree that our data indicate that the high mannose glycans are part of a ‘highly unusual complex’ and must be bound covalently. The referee also agrees with our view that oxidative damage and cross-linking would explain why both tryptic and chymotryptic peptide mapping was unsuccessful in identifying individual peptides. He/she suggests that attempts could be made with alternative proteases, such as GluC. We agree that alternative means of breaking down these complexes could be explored in the future, and indeed, the data shown in Extended Data Fig. 8e and f together with other data we have not included indicate that combinations of proteases may provide a successful strategy. Nevertheless, it is also clear this will be a substantial project requiring additional resources and time beyond the scope of this paper, which focuses on the location and biological effects of the novel high mannose glycans.

Figure 1c: It should be stated that the spectrum is from “one” HbSS donor and, for the sake of the reader, that a ‘normal’ spectrum should be presented alongside it, rather than having the ‘similar’ normal glycome shown in the Extended data. How many normal and ‘abnormal’ glycomes were analysed? If the 48% for the SCD glycome and the 36% for the normal are ‘usual’ percentages (see reply; not mentioned in manuscript), then the higher percentage high mannose in the SCD RBCs becomes significant, but statistics on multiple donors should be included. Also, the question is whether the RBCs were trypsinised before glycan release could be answered in a short statement in the methods; it becomes relevant in the context that tryptic mapping failed to result in identification of the spectrin-associated glycopeptide.

We have clarified that the spectra are from individual HbSS and HbAA donors. We also now present a normal spectrum alongside that from HbSS. Although glycomic analysis by mass spectrometry is only a semi-quantitative technique, we now present the proportions of high mannose glycans from multiple HbAA and HbSS N-glycan spectra pooled from four independent experiments. The mean proportion is higher in HbSS (36.5%) than HbAA (30.0%), but this is not a statistically significant difference (Extended Data Fig. 1e). We have added a line in the methods stating ‘No trypsinization was performed before any glycan analysis’.

Figure 2a and Extended Data 3: State the n number in the legend.

The numbers are now included (note Extended Data Fig. 3 is now Extended Data Fig. 4).

Figure 5: why not have data for HbSS also in panel a?

We now incorporate these data and also include data from HbSC RBCs (note Extended Data Fig. 5 is now Extended Data Fig. 6).

Extended Data 4: include the overall spectrum in Figure 1; it should be stated that these are the glycans from one HbAA donor. The zoom factor for the 'high mass' spectrum should be indicated (also for Figure 1c) so that readers can assess also the low abundance glycans. The annotations differ between the SCD and normal glycomes – e.g., are there branched LacNAc or multiantennary glycans? (Or is there ambiguity/mixture?)

We have now included an HbAA spectrum in Fig. 1 and have clarified that the spectra are from individual donors. New spectra, including zoom factors, have been added (Extended Data Fig. 3), with consistent annotations for both HbAA and HbSS samples.

Extended Data 7: Some major peaks in panel b are not identified – if impurities, then this should be stated.

We now clarify that “Peaks annotated with an asterisk (*) do not correspond to glycan structures. Major structures are annotated for clarity.” (Extended Data Fig. 8c). The spectra from Fig. 1c and Extended Data Fig. 7c are now included in Extended Data File 1, ‘Maldi data.xlsx’.

Reviewer #3 (Remarks to the Author):

The authors provide a revised version of the manuscript that includes a substantial amount of new data on the biochemical characterisation of the mannose-containing ligands exposed in HbSS, oxydised, infected and aged erythrocytes. I still think this is a relevant and important manuscript and, in my view, some of the additional data improve the quality of the work. I strongly sympathise with the research team as they worked very hard to address this challenging problem.

We thank the referee for their appreciation of the work we have carried out over the last 18 months.

First, I wanted to highlight that I found the new proteomics sections and associated figures very difficult to follow. I believe I got a general idea of what has been done and what this means but I am not sure I got all the details right. I would recommend rewriting the text and the inclusion of cartoons describing the experimental process and better annotation of the figures.

We have substantially rewritten the proteomics sections, and also included a cartoon in Supplemental Fig. 8a. We hope that these sections are now easier to understand.

There are instances of mislabelling. In Figure 4 the legend does not seem to correspond to the data shown; section b does not show “Surface GNA lectin staining by flow cytometry (normalized gMFI)(bottom) and 844 corresponding GNA westerns (top) from HbAS and HbSS ghosts. (Right) Spearman’s 845 rank correlations between FACS data and band intensities, both classified ordinally as 846 high, medium or low (n=27 measurements from 22 individuals).”

Figure 4b was in fact correctly labelled, but obviously not explained well. We have therefore changed the original from:

- b) Surface GNA lectin staining by flow cytometry (normalized gMFI)(bottom) and corresponding GNA westerns (top) from HbAS and HbSS ghosts. (Right) Spearman’s rank correlations between FACS data and band intensities, both classified ordinally as high, medium or low (n=27 measurements from 22 individuals).
- c) GNA lectin western blot from healthy (HbAA) and sickle (HbSS) ghosts

to:

- b) Above are shown further GNA lectin western blots from HbAA and HBSS ghosts. The histogram below the blot shows the flow cytometrically measured surface GNA lectin staining values of the RBCs used to make the ghosts, with each bar corresponding to the cells used to make the western lane above. The r value to the right of the 100kDa size label is Spearman’s rank correlation coefficient between GNA lectin staining values and band intensities, both classified ordinally as high, medium or low (n=27 measurements from 22 individuals). None of the other bands yielded significant correlation coefficients.

Other points are:

Based on the new findings, proteolysis of spectrin in response to stress signals generates low molecular weight bands that display the sugars. Based on this I would expect that pull down of GNA-binding material from HbSS ghosts would show low molecular weight bands recognised by the anti-spectrin antibody. This would agree with the western lot data from Extended Figure 6.

We agree with this interpretation.

Figure 4e includes a pull down of GNA-binding material from healthy erythrocytes and shows a single high molecular weight band recognised by the anti-spectrin antibody which indicates that spectrin might be part of a complex that contains high mannose structures in healthy cells. Based on Figure 4f, these structures are present intracellularly in HbAA red blood cells but are more abundant and closer to the surface in HbSS red blood cells. Based on the images, it seems that the cellular distribution of spectrin differs between HbAA and HbSS red. Is this a possibility? 3D high resolution microscopy results (Figure 4j) are consistent with spectrin being associated with high mannose patches displayed at the cell surface in HbSS erythrocytes.

As seen in Figure 4, the distribution of spectrin in HbSS cells indeed differs in several ways from the uniform pattern seen in HbAA cells. Perhaps this not surprising: sickle cell disease is named after the pathological shapes of RBCs seen in the disease and the shape of cells is determined by their underlying cytoskeletons, of which spectrin is the major component. However, the precise structural basis of high mannose glycan structures remains to be determined and we think it is premature to state that ‘these structures are present intracellularly in HbAA cells’. Although this is possible, they may also represent structural changes of membrane proteins in response to oxidative stress.

I was very intrigued by the formation of high molecular complexes that include spectrin. This was discussed in the manuscript but going through the literature I found the manuscripts below that describe cross-linking of spectrin and haemoglobin linked to senescence and oxidative damage and I wonder if the authors, by limiting the analysis to material that enters a gel, might be missing some important players and if they could detect some haemoglobin in their preparations.

Irreversible spectrin-haemoglobin crosslinking in vivo: a marker for red cell senescence (1983)

L. M. Snyder L. Leb J. Piotrowski N. Sauberman S. C. Liu N. L. Fortier
<https://doi.org/10.1111/j.1365-2141.1983.tb02038.x>

Effect of hydrogen peroxide exposure on normal human erythrocyte deformability, morphology, surface characteristics, and spectrin-hemoglobin cross-linking.
L M Snyder, ... , S Shoet, N Mohandas
J Clin Invest. 1985;76(5):1971-1977. <https://doi.org/10.1172/JCI112196>.

We thank the referee for bringing these papers to our attention and now cite them, in addition to that of Kriebardis *et al.* describing related work that we originally cited. We agree that glycoprotein complexes germane to the processes we have discovered may well not enter gels. We partly addressed this by incubating whole RBC ghosts with trypsin, then analysing the resultant peptides on gels. Indeed, this was part of the work that made us realise that high mannose glycans were associated with high molecular weight aggregates. We hope that future analysis of this previously poorly characterised class of molecules will provide insights into our understanding of oxidation and aging processes.

Reviewers' Comments:

Reviewer #2:

Remarks to the Author:

Cao et al, 3rd version

I will not repeat my previous comments regarding the potential interest in the work.

The manuscript is improved, has some rearranged figures and proves less difficult to read – but it is still full of abbreviations, partly explained, partly acceptable, partly unnecessary (but does one really need to abbreviate 'red blood cells' or 'sickle cell trait'?), partly whose meaning is to be guessed (HbAA, HbSS, HbAS, HbF; I assume homo/heterozygosity) – perhaps it is complicated and clear to the authors, but Nature Communications is meant for a wider audience. It is a pity that the exact nature of the spectrin-associated glycopeptides has not been resolved.

Heading of section, line 190: The low molecular weight complexes are 'relatively' protease resistant, as eventually some digestion occurs.

The glycan annotations on the spectra in Figure 1 and Extended Data Figure 3 are not consistent. For the HbAA sample (Figure 1), more glycans are 'uncertain' as compared to the HbSS sample with more indicated as bisected – however, this differs between Figure 1 and Extended Data Figure 3 – are these the same donors? For example, m/z 2489 is shown in the supplement as bisected for HbAA and HbSS, whereas in Figure 1 of the main text, this glycan is shown with a 'bracket' for HbAA. If all were fragmented, then is it the case that in HbAA less can be defined as bisected, whereas in HbSS this is certain? I assume the HbAA spectrum shown is from a blood group A individual, as judged by the annotations. The legend to Extended 3 should mention the red boxes.

Figure 4 h – this is not clear – if it's not a spectrum with m/z, then it's the sequence in terms of amino acids (and the x-axis should be annotated as such).

The authors should check whether all panels are well visible as some are rather small (Extended 1 d/e scatter plots; Extended 7, surface GNA bar chart; spectrum in Extended 8c).

Reviewer #3:

Remarks to the Author:

The authors have addressed my queries and I found the proteomic analysis much easier to follow.

Luisa Martinez-Pomares

REVIEWER COMMENTS

Reviewer #2 (Remarks to the Author):

Cao et al, 3rd version

I will not repeat my previous comments regarding the potential interest in the work.

The manuscript is improved, has some rearranged figures and proves less difficult to read – but it is still full of abbreviations, partly explained, partly acceptable, partly unnecessary (but does one really need to abbreviate ‘red blood cells’ or ‘sickle cell trait’?), partly whose meaning is to be guessed (HbAA, HbSS, HbAS, HbF; I assume homo/heterozygosity) – perhaps it is complicated and clear to the authors, but Nature Communications is meant for a wider audience. It is a pity that the exact nature of the spectrin-associated glycopeptides has not been resolved.

Our original motivation for using some of these abbreviations was to minimise the number of words, mainly in the abstract. We have now reworded the abstract and no longer use SCD / SCT to indicate sickle cell disease / sickle cell trait in the text, except to indicate labels in figures. However, use of RBC as an abbreviation for red blood cells is so widespread (see for instance its Wikipedia entry) that we consider its use legitimate. We have also added several explanatory sentences and phrases to help readers better understand the nomenclature used to indicate which haemoglobins are present (HbAA, HbSS, HbAS and HbF).

Heading of section, line 190: The low molecular weight complexes are ‘relatively protease resistant, as eventually some digestion occurs.

We have now re-worded the title to use the phrase ‘exhibit protease resistance’ (Line 195-6), which does not imply that this property is absolute.

The glycan annotations on the spectra in Figure 1 and Extended Data Figure 3 are not consistent. For the HbAA sample (Figure 1), more glycans are ‘uncertain’ as compared to the HbSS sample with more indicated as bisected – however, this differs between Figure 1 and Extended Data Figure 3 – are these the same donors? For example, m/z 2489 is shown in the supplement as bisected for HbAA and HbSS, whereas in Figure 1 of the main text, this glycan is shown with a ‘bracket’ for HbAA. If all were fragmented, then is it the case that in HbAA less can be defined as bisected, whereas in HbSS this is certain? I assume the HbAA spectrum shown is from a blood group A individual, as judged by the annotations. The legend to Extended 3 should mention the red boxes.

We are grateful to the reviewer for pointing out inconsistencies in the presentation of glycan spectra, which we have now amended. We can clarify the partial spectra presented in Fig. 1 are extracted from the full spectra presented in Extended Data Fig. 3 and this has been made more explicit in the figure legends. The legend to Extended Data Fig. 3 also now includes references to the red boxes.

Figure 4 h – this is not clear – if it's not a spectrum with m/z , then it's the sequence in terms of amino acids (and the x-axis should be annotated as such).

We have added labelling in Fig. 4h, as well as further explanation in the figure legend, to make it clear the figure is based on the sequence of spectrin, not charge to mass ratios.

The authors should check whether all panels are well visible as some are rather small (Extended 1 d/e scatter plots; Extended 7, surface GNA bar chart; spectrum in Extended 8c).

We have improved the visibility of some of the smaller panels in Extended Data Figs. 1, 7 and 8. as requested.

Reviewer #3 (Remarks to the Author):

The authors have addressed my queries and I found the proteomic analysis much easier to follow.

We thank the reviewer for acknowledging our changes and thank her for her previous comments, which improved the manuscript.